# Reprogramming the antigen specificity of B cells using genome-editing technologies

James E Voss[1,2,3†]*, Alicia Gonzalez-Martin[4†]*, Raiees Andrabi[1,2,3†],
Roberta P Fuller[1,2,3‡], Ben Murrell[5,6‡], Laura E McCoy[7‡], Katelyn Porter[1,2,3‡],
Deli Huang[1], Wenjuan Li[1], Devin Sok[1,2,3], Khoa Le[1,2,3], Bryan Briney[1,2,3],
Morgan Chateau[8], Geoffrey Rogers[8], Lars Hangartner[1], Ann J Feeney[1],
David Nemazee[1], Paula Cannon[8], Dennis R Burton[1,2,3,9]*

[1]Department of Immunology and Microbiology, The Scripps Research Institute, La
Jolla, United States; [2]International AIDS Vaccine Initiative Neutralizing Antibody
Center, The Scripps Research Institute, La Jolla, United States; [3]Scripps Center for
HIV/AIDS Vaccine Immunology and Immunogen Discovery (CHAVI-ID), The Scripps
Research Institute, La Jolla, United States; [4]Department of Biochemistry,
Universidad Autónoma de Madrid (UAM) and Instituto de Investigaciones
Biomédicas Alberto Sols (CSIC-UAM), Madrid, Spain; [5]Department of Medicine,
University of California, San Diego, San Diego, United States; [6]Department of
Microbiology, Tumor and Cell Biology, Karolinska Institutet, Stockholm, Sweden;
[7]Division of Infection and Immunity, University College London, London, United
Kingdom; [8]Department of Molecular Microbiology and Immunology, Keck School of
Medicine, University of Southern California, Los Angeles, United States; [9]Ragon
Institute of Massachusetts General Hospital, Massachusetts Institute of Technology,
and Harvard, Cambridge, United States

*For correspondence:
jvoss@scripps.edu (JEV);
alicia.gonzalezm@uam.es (AG-M);
burton@scripps.edu (DRB)

†These authors contributed
equally to this work
‡These authors also contributed
equally to this work

Competing interests: The
authors declare that no
competing interests exist.

Reviewing editor: Tomohiro
Kurosaki, Osaka University,
Japan

**Abstract** We have developed a method to introduce novel paratopes into the human antibody
repertoire by modifying the immunoglobulin (Ig) genes of mature B cells directly using genome
editing technologies. We used CRISPR-Cas9 in a homology directed repair strategy, to replace the
heavy chain (HC) variable region in B cell lines with that from an HIV broadly neutralizing antibody
(bnAb), PG9. Our strategy is designed to function in cells that have undergone VDJ recombination
using any combination of variable (V), diversity (D) and joining (J) genes. The modified locus
expresses PG9 HC which pairs with native light chains (LCs) resulting in the cell surface expression
of HIV specific B cell receptors (BCRs). Endogenous activation-induced cytidine deaminase (AID) in
engineered cells allowed for Ig class switching and generated BCR variants with improved HIV
neutralizing activity. Thus, BCRs engineered in this way retain the genetic flexibility normally
required for affinity maturation during adaptive immune responses. Peripheral blood derived
primary B cells from three different donors were edited using this strategy. Engineered cells could
bind the PG9 epitope and sequenced mRNA showed PG9 HC transcribed as several different
isotypes after culture with CD40 ligand and IL-4.
DOI: https://doi.org/10.7554/eLife.42995.001

## Introduction

Protective antibodies against some pathogens require features not easily elicited through affinity
maturation from the human antibody repertoire (*Kepler and Wiehe, 2017*). We wanted to add these
features into the repertoire directly by modifying BCRs using genome-editing technologies. The
existence of antibodies with protective paratopes encoded mostly within their HCs

(*Heydarchi et al., 2016*; *Lee et al., 2017*; *Sok et al., 2017*; *Sui et al., 2009*) suggested that it might be possible to achieve this goal through replacement of the recombined HC variable region alone. In order for engineered HCs to then function as desired, they must pair with endogenous LCs and retain their ability to recognize antigen as chimeric cell surface-expressed BCRs (*Feige et al., 2010*).

We used HIV as a model because, while broadly neutralizing antibodies (bnAbs) against this virus are protective (*Pegu et al., 2017*) and their gene sequences have been well defined (*McCoy and Burton, 2017*), they remain exceedingly difficult to elicit by vaccination (*Mascola and Haynes, 2013*). Previous studies have suggested that the breadth and neutralization potency of a number of bnAbs targeting the HIV Envelope glycoprotein (Env) 'V2 apex' region are largely encoded within unusually long HC complementarity-determining region 3 (CDRH3) loops, which form the majority of contacts with Env (*Julien et al., 2013*; *Lee et al., 2017*; *McLellan et al., 2011*; *Pejchal et al., 2010*). We found that the IgG HC from the V2 apex-targeting bnAb PG9 could pair and be secreted with a diversity of lambda (λ) and kappa (k) LCs (*Figure 1—figure supplement 1*) when co-transfected in HEK293 cells. These included a LC endogenous to a well characterized human B cell line in which we wanted to develop BCR engineering strategies, the Ramos (RA 1) Burkitt's lymphoma (*Klein et al., 1975*). Size exclusion chromatography (SEC) profiles and SDS-PAGE gels of these secreted chimeric antibodies were comparable with the normal PG9 HC/LC pair (*Figure 1—figure supplement 2*). Chimeras were evaluated for their ability to neutralize HIV pseudovirus using the TZM-bl assay (*Sarzotti-Kelsoe et al., 2014*). Twelve HIV pseudoviruses representing the global diversity of HIV-1 strains (*deCamp et al., 2014*) were examined along with six viruses known to be highly sensitive to neutralization by PG9 (*Andrabi et al., 2015*). All PG9 chimeric antibodies neutralized one or more of the PG9-sensitive viruses, and most neutralized multiple viruses from different clades in the global panel (*Figure 1*). No chimeric antibody was as broadly neutralizing as the original PG9 HC/LC pair, indicating significant LC-dependent restriction to neutralization breadth for this HIV bnAb. Most chimeras had measurable binding affinity to 5 strains of recombinant soluble HIV envelope native trimers (SOSIPs) (*Andrabi et al., 2015*; *Voss et al., 2017*) by biolayer interferometry (BLI) (*Frenzel and Willbold, 2014*) (*Figure 1—figure supplement 3*). Autoreactivity was detected for about 60% of PG9 chimeras in the HEp-2 assay (*Copple et al., 2012*) as might be expected because of novel HC/LC interfaces in these antibodies (*Figure 1—figure supplement 4*). We opted to move forward with the development of engineering strategies using the PG9 HC paratope in Ramos B cells because, despite a loss of neutralization breadth with some LCs, the PG9 HC pairs well with diverse LCs (including the functional Ramos λLC). Further, these LC chimeras can be readily detected with a variety of available recombinant native trimer probes, giving us a simple and specific method for detecting successfully engineered B cells.

It is possible to replace segments of genomic DNA in eukaryotic cells by introducing double strand DNA (dsDNA) breaks on either side of the segment in the presence of exogenous donor DNA that will be incorporated between these break sites by host cell DNA repair machinery. This replacement DNA must be flanked by regions of sequence homology (HR) to the genome upstream and downstream of the two break sites in order to be incorporated by homology directed repair (HDR) mechanisms (*Doudna and Charpentier, 2014*; *Schwank et al., 2013*). This presented a unique challenge for the development of donor DNA HRs that would be universally present on either side of the VDJ region in polyclonal human B cells. This is because each human B cell undergoes genomic rearrangements during its genesis to assemble the HC VDJ gene from; 1 of at least 55 functional V genes (along with its individual 5' promoter); 1 of 23 functional D genes; and 1 of 6 functional J genes. Together, these gene segments span nearly one megabase (Mb) when in germline configuration and constitute the immunoglobulin heavy chain variable (IGHV) locus on chromosome 14 (at 14q32.33) (*Giudicelli et al., 2005*; *Watson and Breden, 2012*) (*Figure 2A*). To respond to this challenge, we developed a long-range HDR strategy that introduced dsDNA breaks after the most 5' V-gene promoter (V7-81), and after the most distal J gene (J6), because HRs 5' and 3' to these sites respectively are retained in all B cells. The distance between these cut sites will vary depending on which V and J genes were assembled in a given B cell. Thus, regardless of which genes were previously assembled, our editing strategy introduces the new VDJ gene in a way that allows transcription using a naturally regulated V-gene promoter from its native locus where it would be subject to hypermutation by AID. When paired with an endogenous cell LC, a chimeric immunoglobulin (Ig) will be secreted as the isotype determined by the genomic configuration of the HC constant region in the engineered cell.

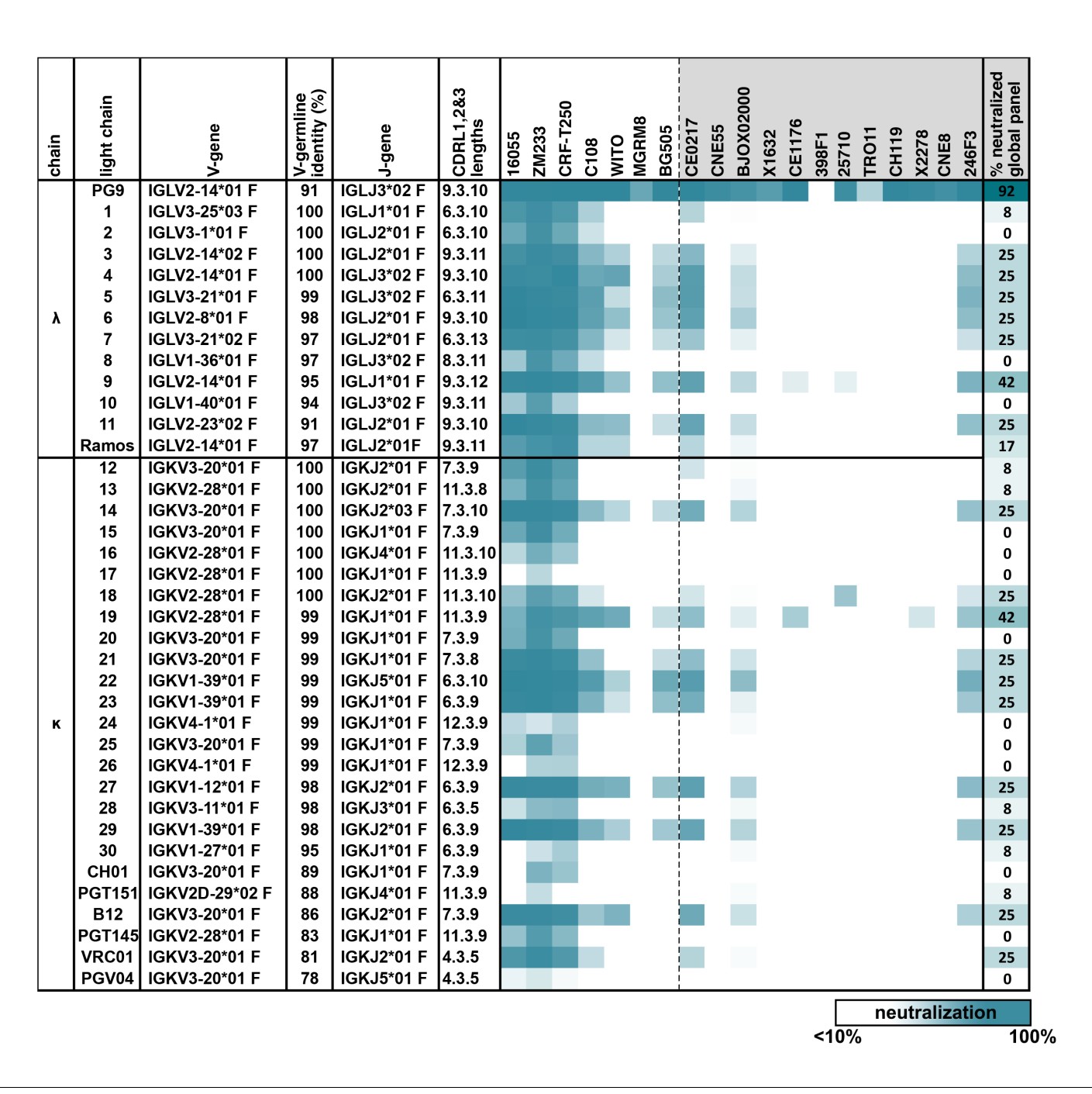

**Figure 1.** PG9 IgG heavy chains neutralize HIV when paired with a diversity of light chains. Sensitivity of 19 HIV isolates to PG9 HC-chimeric LC antibodies are shown in a heat map. Viruses include: six strains especially sensitive to PG9 (leftmost) and 12 viruses representative of the global diversity of HIV (rightmost). The PG9 chimeras are grouped according to lambda and kappa gene usage in order of least to most somatically mutated (amino acid sequences are given in *Figure 1—figure supplement 1*, with the PG9HC/PG9LC control antibody at the top. A diversity of LCs was chosen including several LCs derived from other known HIV bnAbs. LC features are given including IMGT-derived V and J germline gene assignments and sequence identity to the assigned V-gene. CDRL (1, 2 and 3) amino acid lengths are also given. A dark teal to white heat map represents 100% to 10% or less neutralization of the indicated strain of psuedovirus at a concentration of 10 µg/ml of PG9 chimera IgG as described in Materials and methods. The percentage of viruses from the global panel showing at least 10% neutralization for each PG9 chimeric antibody is given on the right.
DOI: https://doi.org/10.7554/eLife.42995.002

The following figure supplements are available for figure 1:

*Figure 1 continued on next page*

*Figure 1 continued*

**Figure supplement 1.** Amino acid alignment of human antibody light chain (LC) variable region sequences expressed as PG9 HC c himeras in 293 cells
.

DOI: https://doi.org/10.7554/eLife.42995.003

**Figure supplement 2.** PG9 chimera purification.

DOI: https://doi.org/10.7554/eLife.42995.004

**Figure supplement 3.** PG9 chimera IgG binding to HIV Envelope (SOSIP) by Biolayer interferometry.

DOI: https://doi.org/10.7554/eLife.42995.005

**Figure supplement 4.** PG9 chimeric IgG autoreactivity test.

DOI: https://doi.org/10.7554/eLife.42995.006

B cell VDJ editing was first performed in the Ramos (RA 1) B cell lymphoma line. This human monoclonal line expresses an Ig HC that uses the IGHV4-34 (V), IGHD3-10 (D) and IGHJ6 (J) genes as IgM (*Borchert et al., 2010*). V4-34 lies halfway through the IGHV locus placing the 5' most V-gene promoter (V7-81) about 0.5 Mb upstream (*Figure 2A*). In addition to our universal B cell editing strategy, which grafts the PG9 VDJ gene between the V7-81 promoter and J6 splice site (*Figure 2B*), we developed an engineering strategy specific for this line that introduced a dsDNA cut 3' of the V4-34 promoter (instead of the V7-81), and which used donor DNA with a V4-34 promoter sequence 5' HR (*Figure 2C*). This strategy replaces only the 400 bp Ramos VDJ rather than a 0.5 Mb region using the 'universal' BCR editing strategy. Cas9 cut sites were identified in the desired regions of the human HC variable locus (red scissors in *Figure 2A*, *Figure 2—figure supplement 1*). Guide RNA sequences synthesized as DNA oligos were cloned into the pX330-SpCas9 and guide RNA expression plasmid using previously published methods (*Ran et al., 2013b*). Cutting activity was assessed for several different guide RNAs targeting each of the three sites using the pCAG-eGxxFP recombination assay in 293 cells (*Mashiko et al., 2013*). Once the highest efficiency cutters were detected using this assay (*Figure 2—figure supplement 1*), corresponding mutations in the PAM sites in donor DNA plasmids were mutated to prevent Cas9 cutting of the donor DNA inside transfected cells (*Figure 2—figure supplement 2*). We introduced either the 'V4-34/Ramos-specific' or the 'V7-81/universal B cell' VDJ editing reagents into cells as two plasmids encoding 5' and 3' dsDNA cutting by Cas9, and one plasmid encoding PG9 VDJ donor DNA (*Figure 2—figure supplement 2*) using nucleofection. Cells were cultured for 3 days to allow for PG9 VDJ gene replacement and HC expression to occur.

To distinguish between the chimeric PG9 HC BCR and the unmodified BCR endogenous to Ramos cells, we used fluorescently labeled soluble HIV envelope trimer (SOSIP) from an isolate of HIV shown to be neutralized by an IgG chimera composed of PG9 HC and Ramos λLC (*Figure 1*), clade AE isolate C108.c03 (*Andrabi et al., 2015*; *Voss et al., 2017*). Cells positive for PG9 HC/Ramos LC chimeric IgM were detected by fluorescence activated cell sorting (FACS) (*Figure 2D*). Cells engineered by the V434 or V781 strategies reproducibly converted an average of 1.75% (SD = 0.20) or 0.21%(SD = 0.03) of transfected cells into Env binding cells, respectively (*Figure 2E*). It was encouraging that the universal editing strategy that removed 0.5 Mb of the IGHV locus was only about 8 times less efficient than the Ramos-specific strategy that replaces just 400 bp. It remains to be determined if this success is particular to the Ig locus in Ramos cells or is a general feature of B cells, which naturally undergo such long distance recombinations during B cell ontogeny (*Ebert et al., 2015*; *Jung et al., 2006*; *Montefiori et al., 2016*). A similar strategy was reported to remove up to 5 Mb of genomic DNA from chromosome 11 in mouse embryonic stem cells suggesting the method may work universally (*Kato et al., 2017*). C108 SOSIP was used to sort successfully engineered HIV-specific cells to produce enriched subpopulations for further experiments. Genomic DNA extracted from these PG9-enriched cells was PCR amplified using primers that annealed upstream and downstream of the expected insertion sites and outside of the donor DNA HRs ('out/out PCR', *Figure 2B, 2F*). Sanger sequencing of these products confirmed that the new PG9 gene was grafted as expected between Cas9 cut sites within the IGHV locus by HDR (*Figure 2—figure supplements 3–5*).

Amplification-free whole genome sequencing of C108 SOSIP-selected cells (engineered with either the V434 or V781 strategy) was performed using fragmented gDNA from these lines using the Illumina HiSeq X. Reads were mapped to the human reference genome hg19 (*International Human*

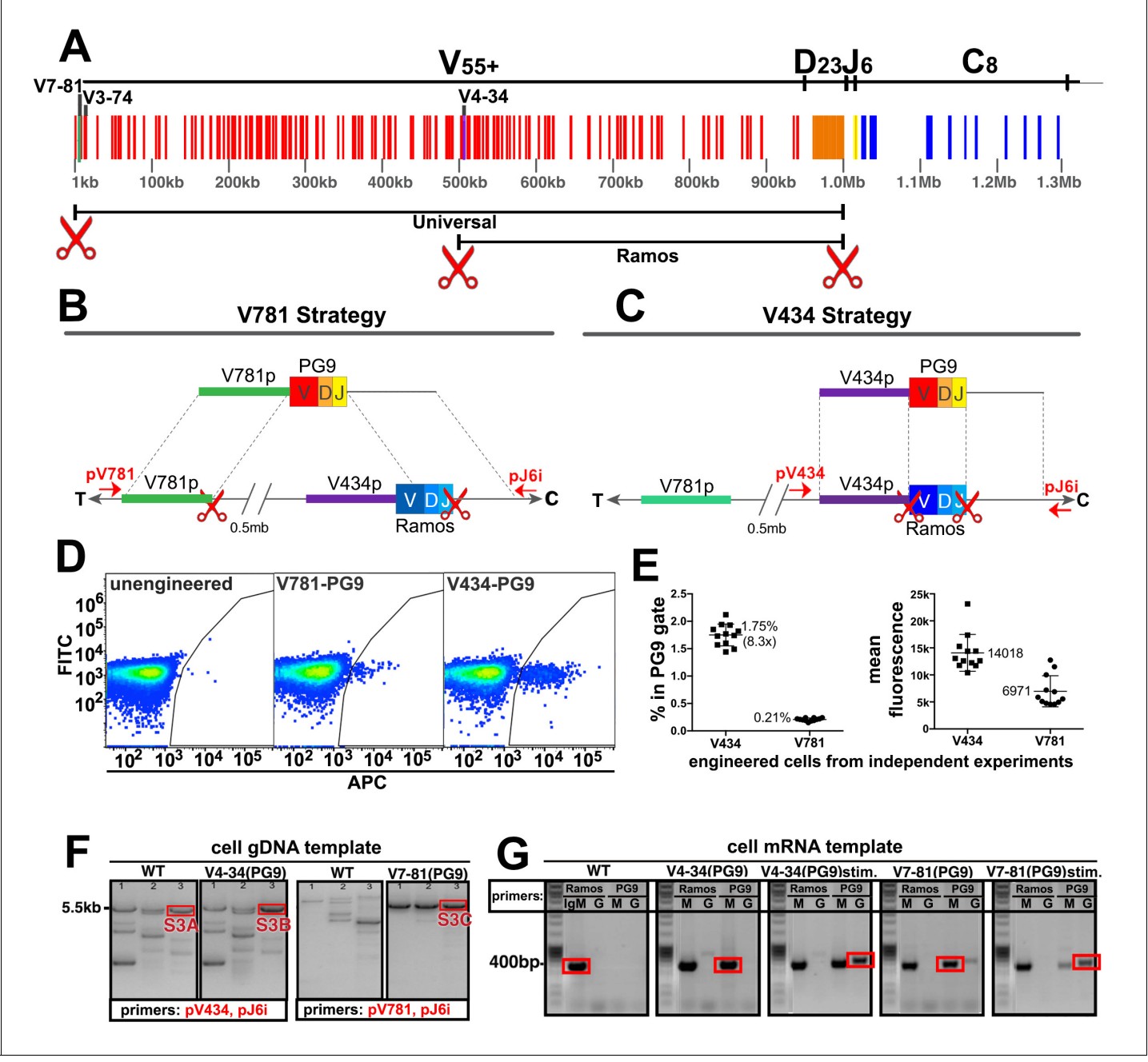

**Figure 2.** Engineering the HC VDJ locus in Ramos B cells. (**A**) Germline configuration of the human immunoglobulin heavy chain locus. The locus is in reverse orientation running 5' to 3' from the telomeric 'T' end towards the centromere 'C' of the long arm of chromosome 14 (14q32.33). It starts with the variable gene region containing at least 55 functional V-genes (in red), 23 functional D-genes (in orange), and six functional J-genes (in yellow) to span almost 1 Mb. The variable region is followed by an intron containing an enhancer element that activates the 5' proximal V-gene promoter in mature B-cells after VDJ recombination. This is followed by the constant gene region (in blue) comprised of 8 functional genes, which is followed by more enhancer elements. The 5' most functional V gene promoter is V7-81 followed by V3-74. The V-gene recombined in the Ramos B cell lymphoma line is V4-34. Scissors represent the location of cas9 dsDNA cut sites developed for BCR editing outlined in this report. (**B**) The universal editing strategy uses cut sites after the V7-81 promoter and J6 gene to replace approximately 0.5 Mb in the Ramos B cell line with PG9 bnAb HC from a donor DNA with HRs upstream and downstream of these cut sites. (**C**) The Ramos specific strategy uses cut sites after the V4-34 promoter and J6 genes to replace only the native Ramos VDJ region (400 bp) with the PG9 VDJ from a donor with HRs upstream and downstream of these cut sites. (**D**) FACS plots of engineered Ramos B-cells (RA1), using either the V781 or V434 HDR strategies. Successfully engineered cells expressing chimeric PG9 BCR bind to APC-labeled recombinant C108 HIV Env trimer (SOSIP). APC positive selection gates were set against the FITC channel to eliminate autofluorescent cells. (**E**) Reproducibility of V781/V434 strategies. Each experiment was reproduced 12 times. The average number of cells able to bind C108 Env

*Figure 2 continued on next page*

*Figure 2 continued*

(SOSIP) after engineering was 0.21% (SD = 0.03) and 1.75% (SD = 0.20) using the V7-81 and V4-34 strategies respectively. Average fluorescence values of APC+ cells from the 12 transfections are also shown. (F) Genomic DNA analysis confirms native VDJ is replaced with PG9 in engineered cells using the V7-81 and V4-34 strategies. PCR reactions were performed on gDNA templates using three primer sets designed to amplify across the entire engineered site including sequence outside of HRs to ensure that new PG9 gene was in the expected context in the engineered cell genomes. Approximate primer annealing sites are indicated by red arrows in *Figure 2B and C*. PCR products using V4-34 promoter/J6 intron primers sets amplified a 5.5 Kb fragment in both V4-34 engineered cells as well as in WT cells (outlined in red rectangular boxes). V781 promoter/J6 intron primer sets amplified a 5.5 Kb fragment in V7-81 engineered cells but not in WT cells. Sequences of PCR products outlined with red boxes are shown in (*Figure 2—figure supplements 3–5*). (G) Engineered cells produce PG9 mRNA transcripts as IgM or as IgG in cytokine-stimulated cells. Ramos 2G6 engineered and C108 SOSIP selected cell mRNA was purified. Primer sets designed to amplify either the wild type or engineered (PG9) heavy chains (IgG or IgM) were used in RT-PCR. Sample template and primers used in the amplification are labelled. Only V4-34 or V7-81 engineered but not WT samples contained PG9-IgM. PG9-IgG could be amplified from CD40L/Il-2/Il-4 stimulated cells. Sanger sequences for the PCR products outlined with red rectangular boxes are given in *Figure 2—figure supplement 10*.

DOI: https://doi.org/10.7554/eLife.42995.007

The following figure supplements are available for figure 2:

**Figure supplement 1.** CRISPR/cas9 guide RNA selection.

DOI: https://doi.org/10.7554/eLife.42995.008

**Figure supplement 2.** Donor DNA nucleotide sequence for the V434 (PG9) 'Ramos specific' and V781 (PG9) 'Universal' B cell engineering strategies.

DOI: https://doi.org/10.7554/eLife.42995.009

**Figure supplement 3.** Amplicon sequence from gDNA derived from wild type Ramos lymphoma B cells.

DOI: https://doi.org/10.7554/eLife.42995.010

**Figure supplement 4.** 'Out/out PCR' amplicon sequence generated from gDNA derived from Ramos B cells engineered using the 'V434' strategy and selected using C108 SOSIP in FACS.

DOI: https://doi.org/10.7554/eLife.42995.011

**Figure supplement 5.** 'Out/out PCR' amplicon sequence generated from gDNA derived from EBV transformed polyclonal cells engineered using the 'V781' strategy and selected using C108 SOSIP in FACS.

DOI: https://doi.org/10.7554/eLife.42995.013

**Figure supplement 6.** Amplification free genome sequencing analysis.

DOI: https://doi.org/10.7554/eLife.42995.014

**Figure supplement 7.** Amplification free genome sequencing analysis.

DOI: https://doi.org/10.7554/eLife.42995.015

**Figure supplement 8.** Amplification free genome sequencing analysis.

DOI: https://doi.org/10.7554/eLife.42995.016

**Figure supplement 9.** Linear diagram schematics showing organization of V781 Donor DNA and target chromosome as well as different possible NHEJ and HR repair events at 5' and 3' crispr/cas9 cut sites.

DOI: https://doi.org/10.7554/eLife.42995.017

**Figure supplement 10.** Sanger sequences of PCR products amplified from the cDNA derived from different.

DOI: https://doi.org/10.7554/eLife.42995.018

*Genome Sequencing Consortium, 2004*) showing approximately 30x coverage depth for diploid regions of the genome (*Figure 2—figure supplement 6*). Coverage depth was consistent with removal of the V781-J6 region in the expressed allele in C108 SOSIP-selected cells engineered with the universal strategy, but not from the excluded allele which is not expressed due to a chromosomal translocation of the IGHV region to chr8 in this Burkitt's lymphoma line (*Klein et al., 1975*). Coverage depth analysis showed a deletion between the V3-11 and V3-7 genes in the excluded allele (relative to the reference genome), and a D-J recombination event in this allele between D3-9 and the J6 gene that would have occurred during the maturation of this B cell in vivo. The high read coverage depth for the 5' and 3' HRs in engineered cells (especially using the V7-81 strategy), suggests the integration of multiple donor DNAs by NHEJ mechanisms (*Figure 2—figure supplement 6*). The presence of donor DNA backbone sequences in this assembly on the outside of HR mapped reads (*Figure 2—figure supplements 7–8*) confirms that integration did occur in both V781 and V434-engineered, C108 SOSIP-selected cells by NHEJ, in addition to the intended HR driven graft of PG9 VDJ into the expected sites. It is normal for DNA introduced as plasmid to be incorporated into the mammalian genome at the site of dsDNA breaks non-specifically by NHEJ (*Vasquez et al., 2001*). The donor DNA format should be adapted to skew repair towards HDR mechanisms if incorporation by NHEJ is to be reduced or eliminated in future experiments. Our design should not allow

for expression of PG9 HC when the donor is incorporated by most off-target repair mechanisms because the PG9 VDJ gene promoter must be in proximity to the downstream enhancer for transcription, and because the gene must be transcribed with the downstream constant gene and spliced in order to make a functional heavy chain protein.

Some predicted, off-target editing events were detected by PCR (*Figure 2—figure supplement 9*) in V781 PG9-engineered unselected or C108 SOSIP-selected cells. Deletions between Cas9 cut sites as have been previously reported to occur (*Kato et al., 2017*) were detected in unselected cells, but not in cells enriched for C108 SOSIP binding. One SOSIP selectable off-target donor DNA incorporation event was detected. This event involved HDR within the 3' HR of the donor DNA, allowing for promoter activity (encoded within the 5'HR of the donor) and transcription with downstream constant regions for expression of functional PG9 HC, but with no accompanying 5' HDR from within the same donor DNA plasmid, indicated by the PCR product shown in *Figure 2—figure supplement 9D*. Integration of donor DNA completely by NHEJ mechanisms was also detected and appeared in both unselected and C108-selected cell lines (*Figure 2—figure supplement 9F*). While evidence for incorporation by NHEJ exists at Cas9 cut sites from these data, it remains to be determined if integration events also occurred elsewhere in the genome.

We then assessed the ability of engineered B cells to undergo two key genome alterations that occur during affinity maturation; class switching and somatic hypermutation. Both of these processes are mediated by AID, which is active in Ramos B cells and is regulated to direct the bulk of its activity within the Ig loci (*Kenter et al., 2016*). It was previously shown that in vitro class switching occurs only in a specific Ramos sub-clone, 2G6 (*Ford et al., 1998*). Thus, we repeated the engineering and selected PG9 HC expressing cells in the 2G6 line. RT-PCR amplification using PG9 forward and IgM or IgG reverse primers generated amplicons which were sequenced to show that the engineered locus successfully transcribed PG9 and spliced it in the correct reading frame to either the native Ramos μ constant gene (PG9-IgM), or after culture with CD40 ligand-expressing feeder cells, IL-2 and IL-4, the native Ramos Y constant gene (PG9-IgG) (*Figure 2G*, *Figure 2—figure supplement 10*). Thus, the engineered locus retained the ability to undergo isotype switching.

While isotype switching is not inducible in Ramos RA1, random somatic hypermutation does occur due to constitutive activity of AID (*Baughn et al., 2011*; *Sale and Neuberger, 1998*). We assessed the ability of edited V781-engineered Ramos RA1 B cells to undergo somatic hypermutation to generate higher affinity variants of the PG9-Ramos chimeric antibody in vitro by repeatedly selecting B cell populations with superior binding to Env using flow cytometry. We chose to use Env SOSIP trimers derived from strains MGRM8 (clade AG) and WITO.4130 (clade B) (*Andrabi et al., 2015*; *Voss et al., 2017*) as sorting probes because these proteins showed relatively weak binding to engineered cells (*Figure 3—figure supplement 1A*), and therefore gave the greatest scope for improvement (*Figure 3A*). Sorts were designed using an anti-human lambda light chain probe (*Figure 3—figure supplement 2*) to eliminate the selection of cells with brighter antigen staining that could result from an upregulation of BCR expression on the cell surface. Three rounds of selection and expansion in culture improved the binding of engineered cell lines to SOSIP probes in FACS experiments (*Figure 3—figure supplement 1B*). mRNA was purified from cell cultures at the time points indicated in *Figure 3A*. Ig variable genes were then amplified from cDNA using gene specific primers as previously described (*Briney et al., 2016*). Libraries were sequenced using the Illumina MiSeq and quality trimmed reads were aligned and filtered to remove all sequences corresponding to the Ramos VDJ or a non-functional lambda chain transcript, confirmed in this study by IMGT analysis (*Ye et al., 2013*) to derive from a VJ recombination event between IGLV2-23 and IGLJ3 on the excluded lambda allele in this cell line. Sequences from different samples were compared to detect the enrichment of coding changes during subsequent rounds of selection that could account for improved binding of the cell lines to HIV Env by FACS. A dominant change within the functional Ramos λLC was increasingly enriched in both MGRM8 and WITO SOSIP-selected lines at an AID hotspot, an S97N substitution within the CDRL3. This mutation predicts a shift in a potential N-linked glycosylation site (PNGS) from N95 to 97. Coding changes resulting in the deletion of this glycan were also observed to a lesser degree (*Figure 3B*). Because no interesting residues in the HC of these lines were likewise selected, we investigated whether the LC changes had a functional effect by expressing the PG9-RamosλLC chimeras as IgG with mutations to either remove or shift the PNGS to 97 as selected (LC S97G or S97N). Antibodies with either of these mutations generally improved affinity for HIV SOSIPs from different clades (*Figure 3C*). Furthermore, these mutations

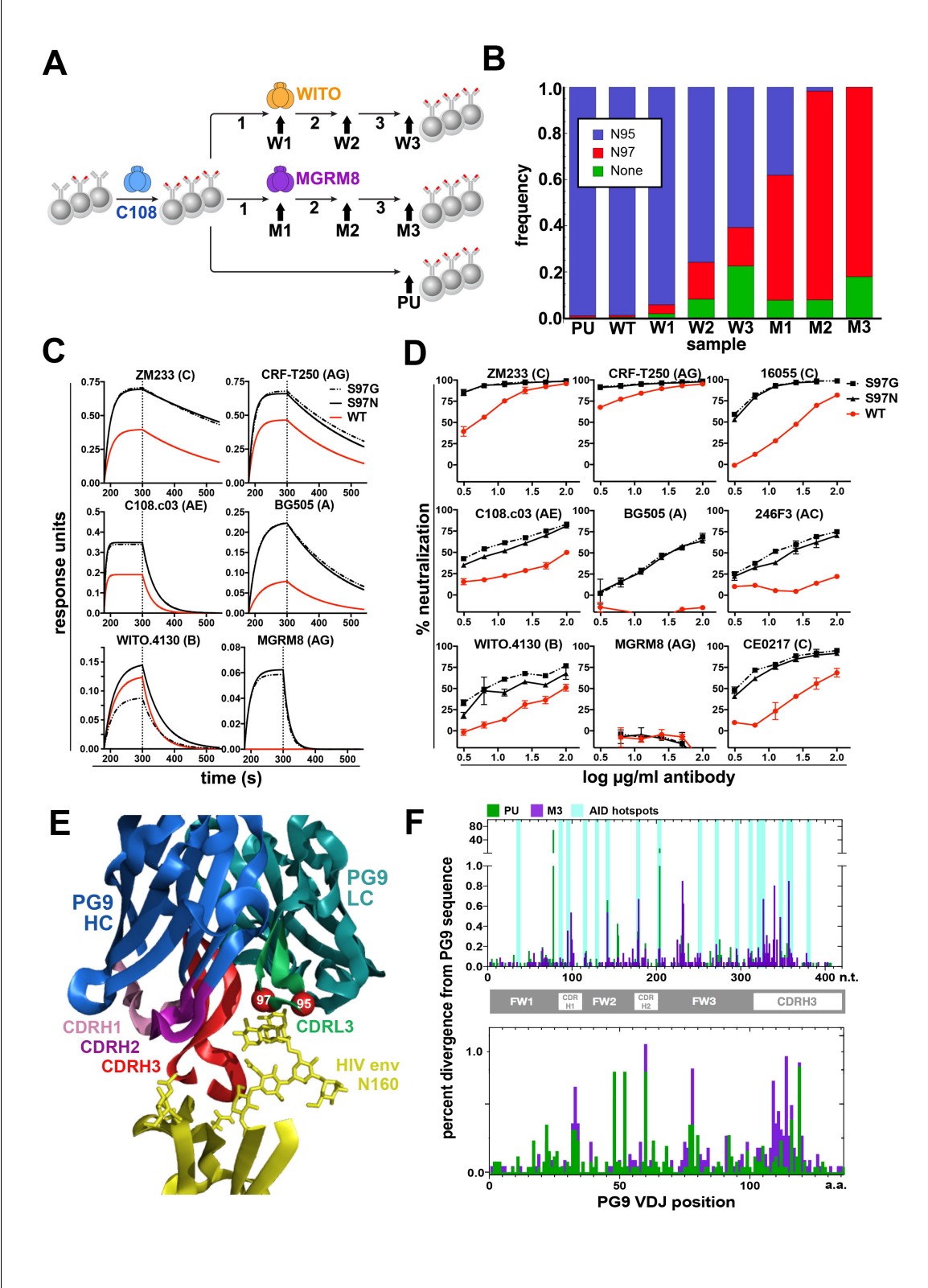

**Figure 3.** Engineered Ramos cells undergo somatic hypermutation and BCR variants with improved HIV neutralizing breadth and potency can be selected. (**A**) Ramos cells engineered to express the PG9 HC VDJ using the V781 universal strategy were enriched using the C108 SOSIP and subsequently passaged without selection (passaged unselected, 'PU') or with three rounds of selection using either WITO or MGRM8 SOSIP strains, (W1-3 and M1-3). gDNA and mRNA samples were obtained for analysis from cultures as indicated by black arrows. (**B**) Ramos LC CDRL3 changes after

*Figure 3 continued on next page*

*Figure 3 continued*

consecutive WITO or MGRM8 SOSIP selection steps. The bar graph shows sequence read frequencies from each sample containing either the original N95, N97, or glycan deleted phenotypes. (C) The wild type PG9HC/RamosLC chimera as well as representative mutants shifting the LC glycan (S97N), or eliminating it (S97G), were expressed as IgGs and characterized for their binding to various HIV Env trimers using Biolayer interferometry (BLI). PG9 chimera-saturated sensors were exposed to 500 mM SOSIP Env trimer (180–250 s) and then PBS (250–500 s) for binding and dissociation kinetics measured as response units (RU). (D) PG9HC/RamosLC WT chimera and the CDRL3 mutant IgGs were tested for neutralization against the panel of pseudoviruses from *Figure 1*. Those showing differences between WT and mutant Abs are shown as neutralization titrations using the TZM-bl assay. % neutralization (y-axis) is shown as a function of IgG concentration (log μg/ml) on the x-axis. (E) The approximate positions of the Ramos CDRL3 residues 95 and 97 are modeled onto the atomic coordinates for PG9 antibody in complex with a HIV Env V2 apex scaffold (PDB:3U2S) and visualized using PyMac. (F) Mutations in the new PG9 VDJ gene in C108 enriched (V7-81 engineered) cells were compared after continuous passage without selection (PU), or with 3 rounds of MGRM8 selection (M3). The nucleotide (top graph) or amino acid (bottom graph) changes from the originally inserted gene are shown in green (PU sample) or purple (M3 sample) by gene position along the x-axis as a percentage of the total analyzed sequences (y-axis). Positions of CDRH1, 2 and 3 are shown in the linear diagram and the locations of AID hotspots are indicated by blue columns on the nucleotide position (x)axis.

DOI: https://doi.org/10.7554/eLife.42995.019

The following figure supplements are available for figure 3:

**Figure supplement 1.** HIV Envelope trimer (SOSIP) binding to engineered Ramos cells after repeated FACS sorts using MGRM8 or WITO SOSIP to enrich higher affinity variants.

DOI: https://doi.org/10.7554/eLife.42995.020

**Figure supplement 2.** Example of gating strategy for the selection of higher affinity B cell receptor variants in V781(PG9)-engineered and C108 SOSIP-enriched cells.

DOI: https://doi.org/10.7554/eLife.42995.021

resulted in more potent neutralization of a number of HIV strains, including a virus from the panel designed to represent global HIV diversity, not neutralized by the original chimera (*Figure 3D*). These mutants remained non-autoreactive in the HeP-2 assay (*Figure 1—figure supplement 4*) and did not strongly alter expression based on SEC profiles (*Figure 1—figure supplement 2*). PG9 crystal structures and homology modeling suggest that, in this chimera, steric clashes involving the Ramos CDRL3 N-linked glycan at position 95 may impede access of the PG9 CDRH3 to its bnAb epitope (*Figure 3E*). This result also suggests that breadth of neutralization by PG9 chimeras may be relatively easy to recover through affinity maturation by a number of possible pathways requiring few mutations to adapt the new light chain.

While interesting coding changes were not significantly enriched in the new PG9 VDJ gene during consecutive rounds of SOSIP selection, we wanted to confirm whether AID was actively mutating this gene over time. The replacement and selection of a defined starting VDJ sequence greatly simplified assessment of mutations introduced at this locus. To eliminate confounding changes in the PG9 VDJ sequence accumulated during RT-PCR amplification, we incorporated unique nucleotide sequence molecular identifiers (UMIs) into the 5' end of each Ig HC mRNA molecule using a barcoded 5' template switch adapter during first strand synthesis by 5' RACE with Ig constant region primers as has been recently described (*Turchaninova et al., 2016*). Illumina adapters were added to PCR amplified libraries for MiSeq sequencing. Consensus sequences derived from reads with identical UMIs (molecular identifier groups, MIGs), were constructed using the previously referenced method. We analyzed only MIGs derived from >29 reads and eliminated unique sequences from the data set to reduce or eliminate chimeric PCR product and any polymerase-introduced mutations from the analysis. We compared data sets coming from two cell lines that had spent the longest time in culture since the original transfection with VDJ gene editing reagents (about 2.5 months). One of these underwent three consecutive selections steps using MGRM8 SOSIP (M3), and one was simply passaged after the initial C108 enrichment and experienced no further selection pressure for SOSIP binding (PU), (*Figure 3A*). Clear differences from the starting PG9 VDJ gene sequences were observed in both data sets and the positions of many of these mutations appear to fall within regions known to be canonical AID hotspots (WRCY) (*Smith et al., 1996*) (*Figure 3F*). This suggests that genes being introduced into the locus undergo normal somatic hypermutation which could be manipulated through the use of codon optimization to encourage or discourage mutations at certain positions by modifying AID hotspot motifs in donor DNA-introduced antibody genes.

We then wanted to test the universal editing strategy in human primary polyclonal B cells that have undergone a diversity of VDJ recombination events, use a variety of different LCs, and which

are readily available in the periphery. We changed the system from one which uses the V7-81 promoter, to one which uses V3-74, (the V gene immediately downstream of V7-81 in the IGHV locus, *Figure 2A*, *Figure 4—figure supplement 1*). In engineered Ramos cells, PG9 expressed using the V3-74 promoter stained more brightly with the HIV envelope probe than did those using V7-81, presumably due to differing promoter strengths. Switching to V3-74 would thus allow for maximal PG9 staining in primary cells which may express lower levels of BCR than the Ramos lymphoma cell line. In addition, engineering efficiency was routinely higher (0.35% of the live cell gate) compared with V7-81 (*Figure 4—figure supplement 1*). PBMCs from different donors were purified by density gradient centrifugation. CD19+ B cells were positively selected by magnetic-activated cell sorting (MACS) and placed into culture with human CD40L and IL-4 to activate and induce cell proliferation. Because cell cycling and/or persistence in the S/G2 phases promote HDR (*Hung et al., 2018*), we stained cells with CFSE to assess the dynamics of B cell division in order to correctly time the introduction of our engineering reagents. Cells were mostly static for the first three days in culture but went through a burst of division by day five under these conditions (*Figure 4A*). We therefore chose to engineer cells on day 4 of culture. We first tested AMAXA nucleofection of a 7.7 Kb GFP plasmid (HR210-PA) into cells at different concentrations (*Figure 4B*). We found that 2 or 3µgs of plasmid per million cells resulted in acceptable levels of transfection allowing us to continue forward testing our engineering reagents in plasmid format. We tested nucleofection of our three-plasmid system at different DNA concentrations and used a 'PG9 chimera' gating strategy (below) to detect successful engineering and expression of the new BCR in cells by FACS at 2, 3 and 6 days post engineering (*Figure 4C*). 2 or 3 µg each of the 5' Cas9, the 3' Cas9, and the V374 donor plasmids/million cells yielded engineering efficiencies approaching 0.1% of the live single cell gate. Because 9 µg of total DNA was significantly toxic to cells reducing their numbers dramatically, we chose to test engineering of B cells from three different donors using 2 µg of each plasmid for a total of 6 µg/million cells (*Figure 4D*). To detect cells expressing PG9 HC as chimeric BCRs with native cell LCs 6 days after engineering, we used ZM233 HIV envelope (in SOSIP format) as a FACS probe, because this strain of virus was observed to be the most broadly neutralized virus in our panel by PG9 chimeras regardless of the LC pair (*Figure 1*). We stained cells using a biotinylated version of this probe bound to either streptavidin-BB515 or streptavidin-APC and selected BB515/APC positive cells to eliminate non-specific or fluorophore-specific binders. We also stained with a version of ZM233 SOSIP (bound to streptavidin-BV421) with a deletion of the N160 glycan, which is a critical component of the PG9 HC binding epitope, in order to gate-out cells binding to other epitopes on the SOSIP. All three engineered donor samples had at least 10-fold more cells in the APC+/BV421-/BB515+ 'PG9 chimeric BCR gate' than did non-engineered controls. After a further 7 days in CD40L/IL-4 culture, cells were harvested, and mRNA was purified (*Figure 4E*). RT-PCR using PG9-specific primers amplified PG9 (confirmed by Sanger sequencing) both as IgM and IgG from engineered cell mRNA (+) but not from nonengineered controls (-) (*Figure 4F*). To quantify PG9 transcripts, RACE-PCR was also performed in order to amplify all transcribed HCs from engineered cell samples (*Turchaninova et al., 2016*). NGS sequencing of these amplicons showed PG9 mRNA-read frequencies similar to the PG9 chimeric BCR-gated cell frequencies observed by FACS seven days earlier (for two of three donors), suggesting survival and similar rates of division for engineered and non-engineered cells in culture. No PG9 mRNA could be detected in non-engineered negative controls. Moreover, PG9 isotypes observed in NGS datasets from engineered cells included IgM, IgD, IgG1 and IgG4 (*Figure 4D*). 3 PG9 MIGs (derived from at least 17 reads) in the donor 1 and 3 error compensated datasets had nucleotide mutations. One of these was a substitution of a C for a T at a predicted AID hot spot motif (WRCY), suggesting the new genes could be undergoing somatic hypermutation as might be expected for primary B cells under these culture conditions (*Johnson et al., 2018*; *Moens and Tangye, 2014*). Despite effective compensation for errors introduced during library preparation by the repertoire analysis methods applied here, there is still a small possibility that these mutations could have arisen during cDNA synthesis, or very early on in PCR amplification of these genes (*Turchaninova et al., 2016*). The AID dependence of observed mutations in engineered genes could be confirmed by comparing these with sequences from engineered cells cotransfected with a guide RNA designed to knock out the AID gene.

This report demonstrates a universal B cell genome editing strategy that introduces novel paratopes into the human antibody repertoire by VDJ replacement in B cells using homology directed repair mechanisms. Using endogenous LCs, engineered cells express immunoglobulins with a

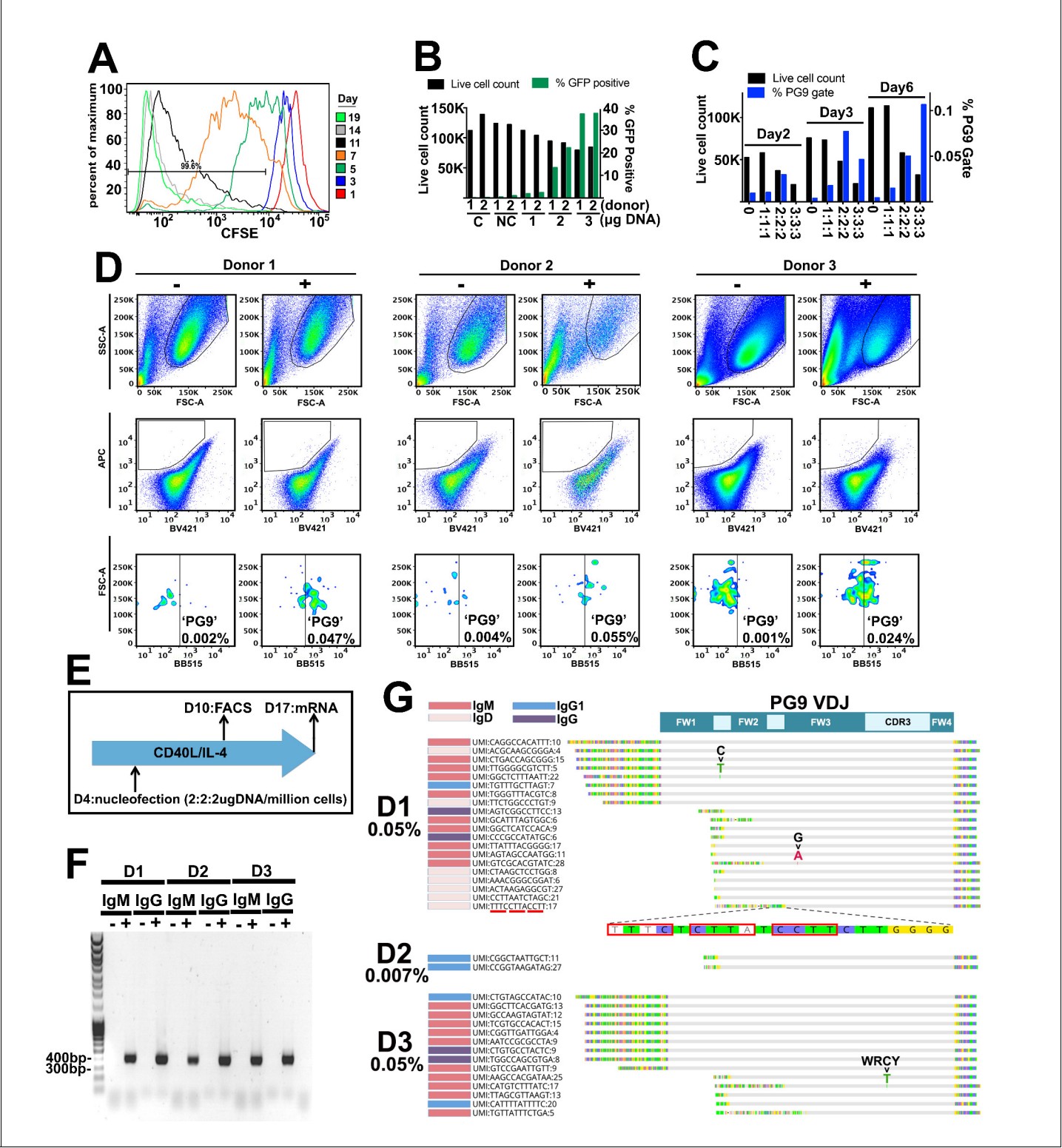

**Figure 4.** Engineering primary human B cells. (**A**) CD19 +B cells were stained with CFSE after purification and cultured in CD40L/IL-4 media. Cells were analyzed for dye brightness on days 1, 3, 5, 7, 11 and 14. Loss of brightness is caused by dilution of the dye that occurs during cell division. Histograms show cell numbers (Y axis) with different CFSE brightness (X axis) on different days from a single donor that was representative of all samples analyzed. (**B**) Cells were nucleofected on day four of culture with a 7.78 kb GFP plasmid. Live cell counts and GFP expression were analyzed 48 hr later and are displayed in the bar graphs. Samples from two different donors were analyzed (1 or 2). C is a no nucleofection control. NC is a no DNA nucleofected control. Other samples were transfected with 1 2 or 3 ug of GFP plasmid as indicated. (**C**) Cells nucleofected on day 4 with engineering reagents were

*Figure 4 continued on next page*

*Figure 4 continued*

stained 2, 3 or 6 days later with ZM233 SOSIP (APC and BB515) and with ZM233 ΔN160 (BV421). Live cell and PG9+ cell gate frequencies for samples nucleofected with 1, 2 or 3 ug of each engineering plasmid are shown. (D) B cells from three different donors were nucleofected on day 4 of culture with 2 ug of each plasmid from the V374 universal B cell editing strategy and stained 6 days later for PG9HC cell surface expression. Frequencies of APC+/BV421-/BB515+ from live single cell gated samples are shown for both engineered (+) and non-engineered (-) controls. (E) Diagram of the engineering and analysis experiment for B cells from three different donors. (F) mRNA from engineered and non-engineered cell cultures was harvested on day 13 post engineering. RT-PCR was performed using PG9 IgM or IgG specific primers. PG9 as both isotypes could be detected in all three engineered donor samples (+) but not in non-engineered controls (-). Sanger sequencing of amplicons confirmed these sequences. (G) Unique molecular identifier (UMI) tagged Ig cDNA libraries were amplified from day 13 mRNA samples using 5'RACE-PCR and sequenced using the MiSeq. PG9 HC was detected in all engineered samples as IgM, IgD, IgG1 and IgG4 isotypes. Frequencies of PG9 reads in these data sets are given. Three molecular identifier groups (MIGs) show mutations in the PG9 VDJ gene as indicated, one of these is in a predicted WRCY AID hotspot motif. N-terminally truncated MIGs from the dataset appear to have template switched prematurely during cDNA synthesis at C-rich regions of the PG9 VDJ gene.

DOI: https://doi.org/10.7554/eLife.42995.022

The following figure supplement is available for figure 4:

**Figure supplement 1.** V3-74 Universal B cell engineering strategy.

DOI: https://doi.org/10.7554/eLife.42995.023

defined specificity, which would be optimally poised to signal antigen-stimulated expansion and affinity maturation in vivo as B cell receptors. The ability to introduce novel paratopes into the human antibody repertoire in this way, could help to surmount difficulties eliciting protective antibody responses against evolved pathogens if it could be done safely and efficiently enough in primary B cells to reproducibly expand protective, self-tolerant antibody responses upon autologous re-engraftment through vaccination. Other reports successfully employing HDR to engineer primary human B cell genomes using ribonucleoproteins (RNPs) and ssDNA donor DNA (nucleofected or AAV6 delivered), suggest this should be possible (*Hung et al., 2018*) (*Johnson et al., 2018*). Further exploration of such a chimeric antigen receptor-B cell (CAR-B) vaccine in animal models should be done in order to test the feasibility of developing this technology for prophylactic or therapeutic use in humans, and as a tool to enhance infectious disease experimental model systems in general.

## Materials and methods

### PG9 chimeric light chain IgG expression

DNA Maxipreps (Qiagen) of human light chain and PG9 heavy chain expression plasmids were sequenced using Sanger sequencing (Eton Biosciences). 25 μg each of filter sterile PG9 heavy chain (IgG) and one light chain antibody expression plasmid (pFUSEss, Invitrogen) from the list of sequences shown in *Figure 1—figure supplement 1*, were co-transfected into 200mls of mycoplasma negative 293 F cells (ThermoFisher Scientific R79007) at $1 \times 10^6$ cells/ml in Freestyle media (ThermoFisher) using PEImax 40K (Polysciences) according to the manufacturer's instructions. Cells were cultured at 37°C in baffled shaker flasks and harvested on day 5 post transfection. The culture was centrifuged, and the supernatant filtered (0.22 μm 500 ml Stericup, ThermoFisher) before IgG purification using a 1:1 mix of PBS washed Protein A/G Sepharose (GE Healthcare). 2 ml of slurry/ 200 ml Ab culture supernatant was used. Briefly, supernatants were loaded overnight onto beads at 4°C. Beads were washed with PBS (250 ml) and eluted with 12 ml 50 mM citric acid buffer pH2.2 into 2 ml neutralization buffer (1M Tris pH 9.0). Eluted IgG was buffer exchanged into PBS using 50 KDa Vivaspin concentrators (Sigma-Aldrich). Each antibody sample was concentrated to 500 μl and purified by size exclusion chromatography (SEC) using an S200 10/30 column (GE Healthcare) in PBS buffer and the 150 KDa peak was pooled and concentrated. IgG concentrations were measured by Nanodrop (ThermoFisher) and stored at 4°C. Non-reducing SDS PAGE gels were run using 5 μg of protein to confirm purity and quality of the IgG produced within one week of purification. Three independently prepared batches of each antibody were generated to confirm reproducibility. SEC profiles and SDS PAGE gels were assessed from each batch to confirm reproducibility. SEC curves and SDS PAGE gels from the third batch (representative of all) are given in *Figure 1—figure supplement 2*.

## Biolayer interferometry

Kinetic measurements were obtained with an Octet Red instrument immobilizing IgGs on PBS hydrated anti-human IgG Fc sensors (Fortebio, Inc). A new sensor was used for each sample. PGT145 purified SOSIP trimers were prepared as previous described (*Voss et al., 2017*) and used as free analytes in solution (PBS, pH 7.4). Briefly, the biosensors were immersed in PBS-T, containing IgG at a concentration of 10 µg/ml for 2 min and at 1,000 rpm prior to the encounter with the analyte. The SOSIP analytes were concentrated to 500 nM in PBS-T. The IgG-immobilized sensor was in contact with the analyte in solution for 120 s at 1,000 rpm and then removed from the analyte solution and placed into PBS for another 250 s. These time intervals generated the association and dissociation binding curves reported in this study. The experiment was repeated with three independent Ig preparations to confirm reproducibility of results. Data from the final reproduction were included in *Figure 1—figure supplement 3*. Analysis was performed within one week of antibody purification.

## Polyreactivity assay: HEp-2 cell staining assay

The HEp-2 cell-staining assay was performed using kits purchased from Aesku Diagnostics (Oakland, CA) according to manufacturer's instructions. These Aesku slides use optimally fixed human epithelial (HEp-2) cells (ATCC) as substrate and affinity purified, FITC-conjugated goat anti-human IgG for the detection. Briefly, 25 µl of 100 µg/ml mAb and controls were incubated on HEp-2 slides in a moist chamber at room temperature for 30 min. Slides were then rinsed and submerged in PBS and 25 µl of FITC-conjugated goat anti-human IgG was immediately applied to each well. Slides were incubated again for 30 min. and washed as above before mounting on coverslips using the provided medium. Slides were viewed at 20x magnification and photographed on an EVOS f1 fluorescence microscope at a 250 ms exposure with 100% intensity. Positive and negative control sera were provided by the vendor. Samples showing fluorescence greater than the negative or PG9 HC/LC control were considered positive for HEp-2 staining. PG9 antibodies were tested within one week after purification.

## Pseudovirus neutralization assays

To produce pseudoviruses, we transfected expression plasmids encoding HIV Envelopes, (12 virus strains from the global panel (*deCamp et al., 2014*) were acquired through the NIH AIDS Reagents Program, others were developed in our laboratory), along with an Env-deficient HIV backbone plasmid (pSG3DEnv) into mycoplasma negative 293 T cells (Sigma-Aldrich 12022001) previously plated with supplemented DMEM in 100 mm tissue culture plates using Fugene (Promega). The cells were cultured for 48 hr before harvesting. 1 ml aliquots of culture were placed in cryovials and immediately stored at −80°C for subsequent neutralization assays.

Pseudovirus TCID50 (50% Tissue Culture Infective Doses) were measured by titrating freshly thawed pseudovirus in mycoplasma negative TZM-bl cells (NIH AIDS Reagent Program 8129). Serial dilutions of pseudovirus in supplemented DMEM were made in white opaque half volume 96 well plates (Corning) (25 µl/well). 25µls of media was added to each well and incubated at 37°C for 1 hr. Then 50µls of media containing $0.5 \times 10^4$ trypsinized cells and 20 µg/ml dextran was added to each well and incubated at 37°C for 48 hr. To assay infectivity, media was removed from the plate and 30 µl/well luciferase lysis buffer (Promega) was added. Bright-Glo Luciferase assay substrate (Promega) was diluted 10X into the lysis buffer (3 ml/plate) and added to the substrate intake of a Bio-Tek Synergy2. The plate reader was programmed to add 30 µl of assay substrate/well before reading luminescence according to the manufacturer's instructions. TCID50 values were calculated using non-linear regression (Prism) from the average of three readings/dilution.

Pseudovirus neutralization by PG9 chimeric antibodies was performed as previously described (*Seaman et al., 2010*; *Walker et al., 2011*). In each well of a white opaque half volume 96 well tissue culture plate, 25 µl of thawed pseudovirus diluted with media (to achieve greater than 100,000 RLUs after 48 hr culture with TZM-bls in the TCID50 Bright-glo assay), was mixed with 25 µl of PG9 chimeric antibody and incubated for 1 hr at 37°C. Then $0.5 \times 10^4$ trypsinized TZM-bl cells in 50µls supplemented media and 20 µg/ml dextran was added and further incubated for 48 hr at 37°C. Neutralization was measured as above using the Bright-Glo assay. In *Figure 1*, rather than reporting neutralization as the half maximal inhibitory concentration (IC50s) from a titration of antibody with

virus, % virus neutralization using 10 µg/ml of chimeric IgG was reported as a heat map (<10–100% neutralization) using the average value from three separate experiments using the different batches of antibodies. Select antibodies were also titrated in a 2-fold serial dilution with media starting at 100 ug/ml and diluting down six wells to obtain the neutralization as a function of Ig concentration (and IC50s derived by fitting data points by non-linear regression using Prism). Antibodies were tested within 1 week of purification.

## B cell engineering reagents

CRISPR/Cas9 guide RNA (gRNA) sequences targeting sites within the IGHV locus of the human reference genome sequence annotated in IMGT (the international ImMunoGeneTics information system) (*Lefranc et al., 2015*), were identified using the Zhang Lab CRISPR design web server (crispr.mit.edu), synthesized as primers (Integrated DNA Technologies, IDT), and cloned into the pX330-U6-Chimeric_BB-CBh-hSpCas9 vector (Addgene plasmid #71707) as previously described (*Ran et al., 2013a*). The top 5 gRNA targets identified were developed for each of the target cut sites; 5' of the V7-81/V3-74 (GenBank: AB019437) and V4-34 (GenBank: AB019439) ORFs; and 3' of the J6 ORF (GenBank: AL122127). Roughly 250 bp sequences containing the CRISPR/cas9 target sequences were either synthesized based on the IMGT annotated reference sequence (Geneart), or PCR amplified from both 293T and Ramos B cell gDNA samples. These targets were cloned into the pCAG-EGxxFP vector (Addgene plasmid #50716), and sanger sequenced (Eton Bioscience). The gRNAs were tested for their efficacy in directing cas9 mediated dsDNA cleavage by co-transfection of the pX330 and corresponding pCAG targets into 293 T cells using PEImax 40K as previously described. Target DNA cutting was scored as GFP expression compared with target DNA only transfected cells because dsDNA cuts within the target will result in HDR that restores the GFP reading frame in the pCAG plasmid as previously described (*Mashiko et al., 2013*). Donor DNA was synthesized as three separate genes (Geneart): The V7-81, V3-74 or V4-34 5' UTR HRs, (5' segment) and the PG9 VDJ ORF and 3' HR (3' segment). A functional PG9 VDJ ORF was designed by grafting the PG9 VDJ nucleotide sequence (GenBank GU272045.1), after the V3-33 start codon, leader peptide, and V-gene intron (GenBank: AB019439), because V3-33 is the germline from which PG9 evolved. The 3' segment (PG9-3'HR) was cloned into the HR110PA-1 donor DNA vector (System Biosciences) using restriction enzymes. Stbl4 electrocompetent bacteria cultured at 30°C was used to propagate the donor DNA vector to prevent modifications in bacteria in HR sequences. The V781, V374 or V434 homology region was then cloned 5' of the PG9 gene by the same methods. PAM sites in CRISPR/cas9 targets within donor DNA homology regions were also mutated using site directed mutagenesis (Agilent Technologies) to prevent donor DNA cutting in B cells also transfected with nuclease plasmids. Donor DNA plasmid sequences were confirmed by Sanger sequencing (Eton Bio). Final nucleotide sequences for donor DNAs are shown in *Figure 2—figure supplement 2* and in *Figure 4—figure supplement 1*.

## Ramos B cell culture and engineering

Ramos (RA I) and (2G6) cells (mycoplasma negative) were obtained from ATCC (CRL-1596 and CRL-1923 respectively) and cultured as directed in IMDM media (Invitrogen) supplemented with 10%FBS, L-glutamine and penicillin/streptomycin. Cells were kept between 0.2 and 2 × 106 cells/ml. Growing cells to high densities disrupted their ability to be transfected. Optimal nucleofection parameters for Ramos RA1or 2G6 lines (from ATCC) were identified using a GFPmax (Lonza) plasmid as described for the Neon transfection System (Life Technologies). Cells were recovered in antibiotic free media and GFP expression was assessed using the Accuri analyzer 48 hr post nucleofection. Settings which gave the highest GFP nucleofection efficiencies (1400V, 20 ms, 2 pulses and 1600V, 20 ms, one pulse for the RA1 and 2G6 subclone respectively), were used to nucleofect 10 µg of HR110PA-1 PG9 donor DNA along with 2.5 µg of each gRNA/Cas9 plasmid (pX330-U6-Chimeric_BB-CBh-hSpCas9) into 5 × 106 cells using the 100 ul tip according to the manufacturer's instructions. Cells were recovered in antibiotic free media and grown for 72 hr (antibiotics were added back 24 hr after nucleofection).

## Engineered ramos cell selection

3–5 days post nucleofection, B cells were washed in PBS and stained in FACS buffer (PBS + 1% FBS) with randomly biotinylated (EZ-Link NHS-Biotin, ThermoFisher), PGT145 purified C108.c03 HIV Env SOSIP (*Voss et al., 2017*) APC labeled streptavidin tetramers (Invitrogen, S32362) as previously described (*McCoy and Burton, 2017*). Briefly, 2 µg of biotinylated SOSIP (1 mg/ml) was mixed with 0.5 µl of streptavidin (1 mg/ml) in 7.5 µl PBS and incubated 30 min at room temperature (RT). 2 µl of this solution was then incubated for 45 min with $5 \times 10^6$ cells in 100 µl FACS buffer. Cells were again washed and single live B cells positive for APC fluorophore were selected using the FACS ARIA III (BD Biosciences) at the TSRI FACS core. Selection gates were made using non-engineered cell controls incubated with the same probe.

## Ramos gDNA sequence analysis

For analysis of PCR amplicons generated from genomic DNA: gDNA was isolated from $3 \times 10^6$ cells using the AllPrep DNA/RNA Mini Kit (Qiagen) for use as template in a PCR reaction using three forward and reverse primer sets specific for genomic regions beyond the 5' and 3' homology regions found within the donor DNAs. These primer sets were designed using the NCBI Primer BLAST server and are listed in *Supplementary file 1* (section I) where 'a' are forward and 'b' are the reverse complement primers of a pair. The reaction was carried out using Phusion HF Polymerase (NEB), 200 ng template, 0.4 µM each primer, 200 µM each dNTP in a total volume of 100 µl. After denaturing at 98°C for 30 s 34 cycles were performed at 98°C for 10 s, 63°C for 30 s, and then 72°C for 3.5 min. followed by a 30 min. hold at 72°C. The 5.5 kb product was purified on 1% agarose and the DNA extracted using the QIAquick Gel Extraction Kit (Qiagen). The PCR product was sequenced using Sanger sequencing (Eton Bioscience) with several primers such that the complete 5.5 kb sequence contig could be assembled (*Figure 2—figure supplements 3–5*). The sequencing primer sequences are listed in *Supplementary file 1* (section I).

Amplification-free whole genome sequencing of V434 or V781 engineered and C108 SOSIP-selected Ramos B cell lines was performed by Genewiz (San Diego, CA). 2 µg of gDNA from either sample was fragmented and Illumina adaptors added using the Trueseq PCR free library prep reagents according to the manufacturer's instructions. Sequencing was performed on Illumina HiSeqX Series with sequencing configuration 2 × 150 PE. HAS 2.0 (Illumina) was used to convert. bcl files into fastq files which were de-multiplexed and adapter sequences removed. Sequence reads were first aligned to the human reference genome (hg19) with the Issac Aligner and Issac Variant Caller (*Raczy et al., 2013*). IGV_2.4.10 was used to prepare figures showing reads that mapped to donor DNA homology regions in this study to show evidence for successful editing of the VDJ region in addition to evidence for other repair events. Individual annotated reads were aligned or nBLASTed to identify specific sequences).

To assess off-target editing events by PCR, genomic DNA was isolated from $4 \times 10^6$ cells using the AllPrep DNA/RNA Mini Kit (Qiagen) for use as template in a PCR reaction. Primers 1–4 (*Supplementary file 1*, section IV) were designed using the NCBI Primer BLAST server. Three primers for each location were tested to select the best for amplification. Primers 5–6 (*Supplementary file 1*, section IV) are standard primers for the donor back bone extended to match Tm of paired primers. The reaction was carried out using Phusion HF Polymerase (NEB), 10 ng template, 0.2 µM each primer, 200 µM each dNTP in a total volume of 25 µl. After denaturing at 98°C for 30 s performed 34 cycles at 98°C for 10 s, 63°C for 30 s, then 72°C for 3 min followed by a 30 min hold at 72°C. The products were visualized on 0.5% agarose.

## Sanger sequencing of ramos ig mRNA

To confirm the presence of PG9 mRNA in Ramos RA one1engineered cells and to detect isotype switching from PG9 IgM to IgG in engineered Ramos 2G6 cells, total RNA was isolated from $3 \times 10^6$ pelleted cells using the AllPrep DNA/RNA Mini Kit (Qiagen) as template for reverse transcription and amplification using the OneStep RT-PCR Kit (Qiagen) with forward primers (Integrated DNA Technologies) specific to the Ramos/PG9 variable regions and reverse primers specific to the IgM/IgG constant regions (*Supplementary file 1*, section II). The reactions contained 400 µM each dNTP, 0.6 µM each forward and reverse primer, 10 ng RNA template, 5U RNasin Plus (Promega) in a total volume of 50 µl. The conditions were 50°C for 30 min, 95°C for 15 min and then 30 cycles of 94°C for

30 s, 58°C for 40 s and 72°C for 60 s followed by an additional 10 min at 72°C. Products were visualized on 1% agarose and purified using the QIAquick PCR Purification Kit (Qiagen). The PCR products were sequenced using Sanger sequencing with the same primers used for the PCR (Eton Bioscience).

## In vitro affinity maturation of ramos lines

Ramos RA 1 cells engineered to replace the endogenous VDJ with PG9 VDJ using the universal (V781) strategy and selected with C108 SOSIP in FACS was passaged 8 times to allow for the introduction of mutations into the Ig variable regions. Cells were titrated with biotinylated PGT145 purified WITO, MGRM8, CRF-T250 or C108 SOSIP, APC-labeled streptavidin tetramers (described above). Cells were incubated with a range of concentrations (3–0.0015 µg/ml SOSIP as tetramer solution) for 45 min in FACS buffer and washed with PBS. APC +gates were set using non-engineered Ramos cells incubated with the highest concentration of SOSIP probe (3 µg/ml SOSIP as tetramer). Engineered cells in the APC +gate at each SOSIP concentration was plotted as a % of total cells against the log of the probe concentration in µg/ml to calculate the effective concentration require to stain 10% of cells (EC10) (*Figure 3—figure supplement 1A*). MGRM8 or WITO probes were incubated with either MGRM8 or WITO APC labeled tetramer as previously described at their EC10 concentrations along with 1000x dilution of anti-human lambda FITC-labeled antibody (Southern Biotech) for 45 min. Cells were washed and live single cells with the highest APC signal (top 5%) after normalization for surface BCR levels (FITC) were selected for subsequent expansion and further sorting with WITO or MGRM8 SOSIPs (an example of which is shown in *Figure 3—figure supplement 2*). This process was repeated twice more with EC10 concentrations for probes calculated before each sort. The starting C108 selected engineered line was also continually passaged throughout the experiment for final mRNA sequencing (PU). At the end of the experiment all cell lines were titrated with C108, CRF-T250, WITO and MGRM8 probes as shown in *Figure 3—figure supplement 1B*. mRNA was harvested from cells after each sorting step and sequenced after RT-PCR/RACE-PCR using next generation sequencing (NGS).

## Next Generation Sequencing of ramos ig mRNA

To characterize the selection of mutations in Ig heavy and light chain variable gene regions in HIV Env SOSIP selected Ramos cells, mRNA was prepared from $3 \times 10^6$ cells (RNEasy kit, Qiagen) and eluted in 50uls of elution buffer. In a DNA clean area, RT-PCR reactions were set up with these mRNA samples using Superscript III (Invitrogen) and gene specific primers (listed in *Supplementary file 1*, section III) according to the manufacturer's instructions. The resulting cDNA was stored at −20°C. Separate heavy and lambda gene PCR amplification reactions were then set up using 1 µl cDNA as template in 25 µl HotStar Taq reactions (Qiagen following) the manufacturer's instructions. Heavy and lambda gene primer (IDT) mixes were made up so reactions would contain 1 µM of each primer in the mix. The thermocycling parameters for this reaction were as follows: 94°C 5 min, (94°C 30 s, 55°C 30 s, 72°C 1 min) x25 cycles, 72°C 10 min, 12°C hold. Second round nested-PCR reactions were performed using Phusion proof reading polymerase (NEB) to amplify and insert priming sites for sample barcoding PCR reactions. Two additional rounds of PCR were performed using universal priming sites added during nested PCR with barcodes specific to the plate number and well location as well as adapters appropriate for sequencing on an Illumina MiSeq. These reactions were performed in a 25 µl volume with HotStar Taq DNA polymerase master mix (Qiagen) according to the manufacturer's instructions. Amplified IgG heavy-and light-chain variable regions were sequenced on an Illumina MiSeq (600-base v3 reagent kit; Illumina). Using the AbStar analysis pipeline (https://github.com/briney/abstar), raw sequencing reads were quality trimmed with Sickle (https://github.com/najoshi/sickle), adapters were removed with cutadapt (https://github.com/marcelm/cutadapt) and paired end reads were merged using PANDAseq (*Masella et al., 2012*). In the case of the Ramos B cell heavy and lambda gene data sets, sequences with less than 90% identity to the PG9 VDJ or consensus expressed Ramos λLC genes were filtered out. Variant counting was performed using Julia and plots made with Mathmatica (*Wolfram research, 2018*).

To characterize the accumulation of mutations in the new PG9 VDJ gene in C108 selected cells which were either passaged for almost 3 months (passaged unselected, PU) or in C108 SOSIP-selected cells which underwent three more rounds of selection using the MGRM8 SOSIP probe (M3),

a second sequencing strategy was used to ensure PCR artifacts from library amplification were eliminated from the analysis. Unique nucleotide sequence molecular identifiers (UMIs) were introduced into each Ig mRNA molecule during first strand synthesis using a barcoded template switch adaptor during 5' RACE with human Ig constant gene reverse compliment primers as previously described (*Turchaninova et al., 2016*). NEBNext Ultra DNA Library Prep Kit adapters were ligated to PCR-amplified libraries and sequenced using 400 × 100 nt asymmetric paired-end reads on Illumina's MiSeq. Data was processed with Migec/Mitools software as described in the above reference. Consensus sequences from MIGs with >29 reads were analyzed. Variant counting was performed using Julia and plots made with Mathmatica (*Wolfram research, 2018*).

## Human primary B cell purification and culture

125 ml blood samples were processed within several hours of being drawn with EDTA to prevent clotting by TSRI normal blood donor services. PBMCs were purified by density gradient centrifugation on Lymphoprep (Alere technologies) in SepMate-50 (Stem cell technologies) tubes. B cells were positively purified from ACK treated, PBS washed PBMCs using human CD19 Microbeads (Miltenyi) according to the manufacturer's instructions. Purified CD19 +B cells were immediately placed into culture at $0.5 \times 10^6$ cells/ml. Primary B cells were cultured in CD40L/IL-4 B cell expansion media (Miltenyi), with antibiotics (pen/strep), according to the manufacturer's instructions. Cells were kept between 0.5 and $1.5 \times 10^6$ cells/ml at 37°C in a humidified incubator with 5.0% $CO_2$. Media was replaced every four days and B cell clusters disrupted by pipetting at the time of media replacement. CFSE staining (CellTrace) was performed according to the manufacturer's instructions.

## Primary B cell nucleofection

After four days of culture, B cells were washed, and B cell clusters disrupted by pipetting up and down in PBS. Cells were pelleted and resuspended in 20 µl of AMAXA P3 nucleofector solution (plus supplement) for every million cells. 1 million cells were mixed with plasmid DNA. The GIGA prepped (Qiagen) plasmids were at high concentrations such that the total DNA volume in EB buffer was less than 2 µl. Cells plus DNA plasmids (or no plasmids in non-engineered negative controls) were placed in cuvettes from the AMAXA V4XP-3032 kit and nucleofected in the 4D-Nucleofector using the E0-117 setting. Cells were immediately added to antibiotic free B cell expansion media at $1 \times 10^6$ cells/ml and cultured. 24 hr after nucleofection an equal volume of B cell expansion media containing 2x pen/strep was added to engineered cells. Media was refreshed every 4 days with washing and disruption of B cell clusters by pipetting in PBS.

## Engineered primary cell FACS staining and sorting

ZM233 HIV envelope based SOSIP recombinant proteins (unpublished data) were used to stain primary B cell cultures engineered to express PG9 HC, with or without a PNGS at residue 160 were coexpressed in 293 F cells with a furin expression plasmid and harvested from supernatants. The version with N160 was purified on PGT145 antibody columns while the N160 deleted variant was purified with lectin and size exclusion chromatography to select the trimeric form as previously described (*Voss et al., 2017*). As with the Ramos cell probe described above, SOSIPs were randomly biotinylated for binding to streptavidin fluorophores. To prepare the positive probes; 1 µg of N160 SOSIP in 5µls of PBS was mixed with either 2.5 µl Streptavidin-BB515 (BD Horizon 564453) or 1 µl of a 2x dilution of streptavidin-APC (Invitrogen S32362) in PBS. To prepare the negative probe, 2 µg of N160 SOSIP in 10 µl PBS was mixed with 10µls of streptavidin-BV421 (BD Horizon 563259). After a 30 min incubation, 5 µl of BB515 probe, 5 µl of APC probe or 10 µl of BV421 probe were each added to separate aliquots of 250 µl of FACS buffer (PBS 1%FBS). The final stain mix was made up by adding 1.125 ml of FACS buffer into a tube and adding 50 µl of the BB515 dilution, 50 µl of the APC dilution and 25 µl of the BV421 dilution. B cells to be stained were washed with PBS and clusters were disrupted by pipetting. Cells between 3 and 7 days post engineering were counted and 100 µl of final stain solution was used to resuspend $1 \times 10^6$ cells. Cells were stained for one hour with rocking at RT and directly diluted with FACS buffer for sorting using the ARIA III (BD Biosciences) or analysis on the ZE5 Cell analyzer (Biorad) at the TSRI FACS core.

## mRNA sequencing of PG9-engineered primary B cells

B cells cultured for 13 days post engineering were harvested and mRNA purified using the RNeasy Micro Kit (Qiagen). Reverse transcription and amplification were done using the OneStep RT-PCR Kit (Qiagen) with forward primers (Eton Bioscience) specific to the Ramos/PG9 variable regions and reverse primers specific to the IgM/IgG constant regions (*Supplementary file 1*, section II). The reactions contained 400 µM each dNTP, 0.6 µM each forward and reverse primer, 10 ng RNA template, 2.5U RNasin Plus (Promega), 0.5 µl OneStep RT-PCR Enzyme Mix in a total volume of 25 µl. The conditions were 50°C for 30 min., 95°C for 15 min. then 30 cycles of 94°C for 30 s, 67°C for 40 s and 72°C for 60 s followed by an additional 10 min. at 72°C. A second PCR was done to amplify the products. These reactions contained 1 µl 1st PCR product, 2X HotStarTaq Plus Master Mix (Qiagen) and 0.5 µM each forward and reverse primer in a total volume of 25 µl. The conditions were 95°C for 5 min. then 24 cycles of 94°C for 30 s, 67°C for 30 s and 68°C for 1 min. followed by an additional 20 min. at 68°C. Products were visualized on 1% agarose.

RACE-PCR was performed using human heavy chain constant primers and a 'smart template switch adaptor' as described in *Turchaninova et al. (2016)*. Libraries were pooled for Illumina sequencing adapter ligation and sequenced using asymmetric 400 × 100 paired end reads on the MiSeq (Illumina) by the TSRI sequencing core as described above for Ramos engineered HC AID mutation analysis. Non-engineered and engineered samples from each of the three donors were sequenced (3 million reads from each sample) and processed using Migec/MiTools. The average MIG size from each dataset ranged from 3 to 11. We accepted MIGs made from three or more reads for analysis. MIGs were processed using Mixcr as described. CDRH3s were mapped to the PG9 CDRH3 reference and these reads extracted for *Figure 4G* Isotype calls were made using Mixcr/ IMGT.

## Acknowledgements

This work was supported by the National Institutes of Health, R01-5R01DE025167-04, by The Bill and Melinda Gates Foundation OPP1183956, by CHAVI-ID grant UM1 AI100663, by the Ramón y Cajal Merit Award from Ministerio de Ciencia, Innovacion y Universidades, Spain (RYC-2016–21155 to AG-M), and by a Marie-Curie Fellowship (FP7-PEOPLE-2013-IOF to LEM). We thank the TSRI Flow Cytometry Core Facility for sorting engineered B cells; Priscilla Crisler at the TSRI Normal Blood Donor Services for blood collection for primary B cell engineering, supported by the Clinical Translational Science Award (CTSA) Grant UL1 TR001114; We thank Steven Head and Jessica Ledesma at the Next Generation Sequencing Core at TSRI for generating MiSeq data sets from UMI labeled cDNA PCR libraries. We thank Christina Corbaci for graphic design work. This is manuscript #29498 from The Scripps Research Institute.

## Additional information

### Funding

| Funder | Grant reference number | Author |
|---|---|---|
| Bill and Melinda Gates Foundation | OPP1183956 | James E Voss |
| National Institutes of Health | 5R01DE025167-05 | Dennis Burton |
| Ministerio de Ciencia, Innovacion y Universidades | Ramón y Cajal Merit Award RYC-2016-21155 | Alicia Gonzalez-Martin |
| FP7 People: Marie-Curie Actions | FP7-PEOPLE-2013-IOF | Laura E McCoy |

The funders had no role in study design, data collection and interpretation, or the decision to submit the work for publication.

## Author contributions

James E Voss, Conceptualization, Resources, Data curation, Formal analysis, Supervision, Funding acquisition, Validation, Investigation, Visualization, Methodology, Writing—original draft, Project administration, Writing—review and editing; Alicia Gonzalez-Martin, Validation, Investigation, Methodology, Writing—original draft, Writing—review and editing; Raiees Andrabi, Conceptualization, Supervision, Investigation, Methodology, Writing—review and editing; Roberta P Fuller, Validation, Investigation, Visualization; Ben Murrell, Software, Formal analysis, Visualization; Laura E McCoy, Conceptualization, Resources, Writing—review and editing; Katelyn Porter, Resources; Deli Huang, Methodology; Wenjuan Li, Khoa Le, Investigation; Devin Sok, Supervision; Bryan Briney, Software; Morgan Chateau, Geoffrey Rogers, Lars Hangartner, Writing—review and editing; Ann J Feeney, David Nemazee, Resources, Methodology, Writing—review and editing; Paula Cannon, Funding acquisition, Writing—review and editing; Dennis R Burton, Funding acquisition, Writing—original draft, Writing—review and editing

## Author ORCIDs

James E Voss (iD) http://orcid.org/0000-0002-4777-1596

## Decision letter and Author response

Decision letter https://doi.org/10.7554/eLife.42995.032
Author response https://doi.org/10.7554/eLife.42995.033

# Additional files

## Supplementary files

• Supplementary file 1. List of primers used to generate PCR products presented in this study.
DOI: https://doi.org/10.7554/eLife.42995.024

• Transparent reporting form
DOI: https://doi.org/10.7554/eLife.42995.025

## Data availability

Next generation sequencing data from RT-PCR amplicons have been deposited at Dryad: DOI: https://doi.org/10.5061/dryad.45j0r70. Amplification free whole genome sequencing reads mapped to the human reference genome have been deposited to NCBI with BioSample accession numbers SAMN09404498 and SAMN09404497

The following datasets were generated:

| Author(s) | Year | Dataset title | Dataset URL | Database and Identifier |
|---|---|---|---|---|
| James E Voss, Alicia Gonzalez-Martin, Raiees Andrabi, Roberta P Fuller, Ben Murrell, Laura E McCoy, Katelyn Porter, Deli Huang, Wenjuan Li, Devin Sok, Khoa Le, Bryan Briney, Morgan Chateau, Geoffrey Rogers, Lars Hangartner, Ann J Feeney, David Nemazee, Paula Cannon, Dennis R Burton | 2018 | Data from: Reprogramming the antigen specificity of B cells using genome-editing technologiesDOI: | https://doi.org/10.5061/dryad.45j0r70 | Dryad, 10.5061/dryad.45j0r70 |
| James E Voss, Alicia Gonzalez-Martin, Raiees Andrabi, Roberta P Fuller, Ben Murrell, Laura E McCoy, Katelyn | 2018 | PG9HC(V434)Ramos-WGS | https://www.ncbi.nlm.nih.gov/sra?LinkName=biosample_sra&from_uid=9404498 | NCBI Sequence Read Archive, SAMN09404498 |

Porter, Deli Huang

| James E Voss, Alicia Gonzalez-Martin, Raiees Andrabi, Roberta P Fuller, Ben Murrell, Laura E McCoy, Katelyn Porter, Deli Huang, Wenjuan Li, Devin Sok, Khoa Le, Bryan Briney, Morgan Chateau | 2018 | PG9HC(V781)Ramos-WGS | https://www.ncbi.nlm.nih.gov/sra?LinkName=biosample_sra&from_uid=9404497 | NCBI Sequence Read Archive, SAMN09404497 |
| --- | --- | --- | --- | --- |

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
