## [Decision Letter]

[Editors’ note: a previous version of this study was rejected after peer review, but the authors submitted for reconsideration, and the paper was reviewed and accepted. The original decision letter after peer review is shown below.]

Thank you for submitting your work entitled "Reprogramming the antigen specificity of B cells using genome-editing technologies" for consideration by *eLife*. Your article has been reviewed by three peer reviewers, and the evaluation has been overseen by a Reviewing Editor and a Senior Editor.

Our decision has been reached after consultation between the reviewers. Based on these discussions and the individual reviews below, we regret to inform you that your work will not be considered further for publication in *eLife*.

As seen in the reviewers' comments, they feel that this is an interesting study in terms of polishing up the CRISPR/cas method for future advances in this field.

However, the weak point of this manuscript is to employ the B cell lines. Instead, it would reach the *eLife* level, if authors could demonstrate this utility by using more in vivo situations such as hematopoietic lines that give rise to mature lymphocytes or growing untransformed B cell lineage cells. Since it would take more than two months, because of the journal policy, we would reject this manuscript at this moment.

*Reviewer #1:*

The manuscript by Voss et al. represents an interesting, albeit modest technical advance, in the use of CRISPR/cas-9 technology to insert novel VDJ rearrangements into the endogenous loci of genetically diverse B lymphocytes. The advance comes with a modestly efficient strategy to target the desired insert into some substantial number of different VDJ joins that arise naturally by chance. There are obvious and substantial possibilities for this technology. For example, although it is unlikely to be utilized in human therapy (CRISPR/cas off-target effects are now known to be substantial) one could imagine using this approach in non-human primates to great effect in studies of protection and/or virus selection. Such an experimental model would be broadly significant and a substantial boon to infectious disease research.

The weakness of this manuscript is that it does not really approach this interesting goal. Instead the genetic work is carried out in B-cell lines and EBV transformants. The observation that the VDJ replacement in situ is expressed and subject to AID activity is expected and rests on a substantial body of prior work.

That it is possible to insert/modify endogenous gene loci by CRISPR/cas in murine hematopoietic cell lines that give rise to mature lymphocytes in vivo has been well shown by the Rajewsky laboratory. If Voss and colleagues had done something similar in human (D. Baltimore has described this type of ex vivo development for generating mature human B cells) or macaque progenitors, this work would have considerably more impact.

*Reviewer #2:*

In the manuscript "Reprogramming the antigen specificity of B cells using genome-editing technologies", James Voss et al. developed new technology replacing Ig heavy chain (HC) variable region to desired Paratope sequence in immortalized B cells. By targeting productive VDJ to endogenous genomic Ig locus of B cells, their method allows endogenous control of targeted Ig HC at expression level, light chain paring, somatic mutation, and isotype switching, potentially without damaging genome elsewhere. Interestingly, targeting is designed to enable VDJ replacement in any type of B cells and targeted BCR variable region can be further modified by in vitro affinity based selection system. Present engineering work based on cell lines will provide useful information for future advance in this field, i.e. BCR genome engineering of primary B cells aiming transplantation therapy or vaccination.

[Minor comments not shown.]

*Reviewer #3:*

The manuscript by Voss et al. describes the use of CRISPR-cas9 to engineer Ramos cells, and later EBV transformed B cells to express the heavy chain of an HIV-directed bNAb, PG9. Pairing of this mAb with the endogenous Ramos light chain resulted in functional antibodies with some degree of breadth, a reflection of the H3-dominant mode of binding of this antibody, and the authors show evidence of class-switching and somatic hypermutation. The authors speculate that this may form the future basis of human genome editing, though more immediately this approach may be valuable in the evaluation of immunogens.

Main text, second paragraph, Figure 1 – the loss of breadth for the PG9 heavy chain with any of these light chains is greater than implied by the authors. Adding percentage PG9 breadth to Figure 1 may make this clearer. (It is intriguing that 3 members of the global panel were fairly consistently neutralized by virtually all chimeras).

Figure 2 – the legend for parts B and C seems to have been reversed. Also, the text describes the selection of engineered cells using SOSIP trimers labelled with APC, however the axes for Figure 2D indicate a FITC labelled trimer was used (or the axes are mislabeled). Finally, in panel G, there seems to be evidence of some class switching in the unstimulated V781 cells. The labelling of panel G is cumbersome and could be simplified to make this easier to interpret.

Main text, fourth paragraph – is CRISPR-cas editing efficiency generally known to be related to the size of the segment being replaced?

Main text, ninth paragraph and Figure 3B. Was equivalent sequencing performed for the light chains of cells that were not exposed to selection by WITO and MGRM8? This should be shown for comparison with W1,2,3 and M1,2,3. Also do positions 95 and 97 fall within AID hotspots?

Figure 3B does not make it easy to see which of the mutations result in the shift or deletion of the glycan at position 95 which is the focus of the following section.

Main text ninth paragraph. The authors state that antibodies with the N95 glycan deleted or shifted to 97 showed improved affinity for HIV SOSIPS. While this seems generally true, it is not true for WITO, one of the selecting antigens, where loss of the glycan results in a drop in affinity. Since WITO-selected cells lose the glycan as often as they shift it, how do the authors explain the selection of deleterious mutations at this level?

Figure 3E – how similar are the Ramos and PG9 light chains? Would homology modelling with the correct light chain provide additional confirmation?

Figure 3F – related to the comment above, a similar comparison between PU, M3 and W3 would be informative, given the substantial SHM in the H chain in PU, some of which overlaps that of the antigen selected SHM in M3.

[Minor comments not shown.]

---

## [Author Response]

We recently submitted our manuscript to be reviewed for publication in *eLife*. Two of three reviewers were very positive about publishing the study because it is the first to describe a successful genome editing approach for introducing novel antibody paratopes into the human repertoire through B cell receptor (BCR) modification in polyclonal human B cells. Introducing new paratopes in such a way could aid in the development of vaccines against evolved pathogens for which failed efforts are mostly the result of genetic limitations imposed by the human antibody repertoire itself. The study uses HIV as an example, because while protective broadly neutralizing antibodies to this virus do exist and have been well characterized, they are extremely difficult to elicit through vaccination because they derive from rare precursor BCRs which further require extensive somatic hypermutation. Here, we show it may be possible to bypass these bottlenecks by introducing the protective paratope from an HIV broadly neutralizing antibody (PG9) directly into the repertoire using genome editing technologies.

One reviewer felt that our system, developed in cell lines, should be shown to function in human primary B cells in order to complete this study for publication in eLife. We agree that including such a result would dramatically improve the impact of this report. We therefore have replaced data showing success engineering in polyclonal EBV transformed human B cells, with data showing similar success in human primary B cells purified from fresh PMBCs. Here, activated cells were engineered using the plasmid reagents developed in cell lines. 6 days after engineering, healthy live single B cells could be observed to bind the PG9 epitope of HIV envelope probes by FACS. 13 days after engineering and continuous passage in culture, mRNA was harvested and characterized. PG9 transcripts could be detected in engineered cell cultures at frequencies consistent with PG9 epitope binding frequencies observed by FACS. Moreover, these transcripts were expressed as several different isotypes and were accumulating mutations in the VDJ region consistent with AID activation by IL-4 in this culture condition. In this version of the manuscript we have also addressed all of the other reviewer´s concerns.

Reviewer #1:

[…] The weakness of this manuscript is that it does not really approach this interesting goal. Instead the genetic work is carried out in B-cell lines and EBV transformants. The observation that the VDJ replacement in situ is expressed and subject to AID activity is expected and rests on a substantial body of prior work.

*That it is possible to insert/modify endogenous gene loci by CRISPR/cas in murine hematopoietic cell lines that give rise to mature lymphocytes* in vivo *has been well shown by the Rajewsky laboratory. If Voss and colleagues had done something similar in human (D. Baltimore has described this type of* ex vivo *development for generating mature human B cells) or macaque progenitors, this work would have considerably more impact.*

We take the reviewer’s points but believe that the study is valuable to establish a strategy for introducing useful BCR specificities into polyclonal B cells and that this is a prerequisite for exploration of the idea of establishing the technology for prophylactic or therapeutic application in people. CRISPR/Cas9 and other nucleases are continually being evolved for improved cutting efficiency and elimination of off-target effects. In addition, other nuclease free methods exist for initiating homology directed repair in cells using single stranded donor DNA (German G Gornalusse et al., Nature Biotechnology 15 May 2017). We agree that in the shorter term, this strategy may prove most useful in various experimental models which we now specifically discuss in the conclusions. In this study, we chose to target mature B cells rather than hematopoietic stem cells for several reasons. Because engineered cells must eventually be autologously engrafted in most animal models, using substrate cells readily obtained in the periphery would make eventual re-engraftment easier. Additionally, we might not need high engineering efficiencies in mature B cells compared with that which might be required for progenitor cells because we can exploit their natural potential for antigen dependent expansion and maturation in vivo through the use of immunogens during engraftment. We agree that engineering primary B cells rather than B cells adapted for growth in culture would make the study significantly more impactful. We have therefore replaced Figure 4 showing engineering success in polyclonal EBV transformed cells with results in human primary B cells purified from fresh PBMCs. NGS mRNA sequence datasets from these experiments will be uploaded to Dryad and NCBI for review.

Reviewer #3:

[…] Main text, second paragraph, Figure 1 – the loss of breadth for the PG9 heavy chain with any of these light chains is greater than implied by the authors. Adding percentage PG9 breadth to Figure 1 may make this clearer. (It is intriguing that 3 members of the global panel were fairly consistently neutralized by virtually all chimeras).

We have now changed the text to say ‘No chimeric antibody was as broadly neutralizing as the original PG9 HC/LC pair, indicating *significant* (instead of ‘some’) LC-dependent restriction to neutralization breadth’. We have also added percentage values for the number of viruses from the global panel showing at least some neutralization sensitivity to the indicated PG9 chimera. We agree with the reviewer that it is intriguing that 3 viruses from the global panel are more ubiquitously neutralized by PG9 chimeras. Indeed, we looked into this and there does not seem to be an obvious common feature shared between these three viruses compared with the other nine based on sequence alignments of the V1/V2 apex epitope region (Figure 9, DeCamp et al., 2014).

Figure 2 – the legend for parts B and C seems to have been reversed. Also, the text describes the selection of engineered cells using SOSIP trimers labelled with APC, however the axes for Figure 2D indicate a FITC labelled trimer was used (or the axes are mislabeled). Finally, in panel G, there seems to be evidence of some class switching in the unstimulated V781 cells. The labelling of panel G is cumbersome and could be simplified to make this easier to interpret.

We thank the reviewer for pointing out these mistakes and have fixed them in the revised manuscript. The X axis in Figure 2D is now correctly labeled as ‘APC’. We were also curious about the detection of what looked to be PG9 IgG in unstimulated engineered (C108 selected) V781PG9 HC-expressing cells. Sanger sequencing indeed confirmed that this band was PG9 IgG and not an off-target product amplified during PCR. Because no IgG bands could be detected in any other unstimulated sample, we wonder if we may have selected a 2G6 sub-clone during the V781 engineering bottleneck which had the ability to class-switch in culture without cytokine stimulation, perhaps at low levels, such that class switching in the minority (Ramos) heavy chain in this C108-selected line could not be seen by PCR amplification of IgG mRNA. We think that the low level of Ramos HC mRNA in engineered, selected cell lines is the reason we do not see amplification of Ramos HC IgG in stimulated cultures even though it can be detected as IgM. We have modified Figure 2G to make the labelling less cumbersome.

Main text, fourth paragraph – is CRISPR-cas editing efficiency generally known to be related to the size of the segment being replaced?

In most genome editing studies, employing two Cas9 cut sites, the desired outcome is a deletion of the region between these sites using NHEJ mechanisms. It has been observed in several reports (referenced in Yuning Song et al. Oncotarget v.8(4):5647, 2017), that this outcome is observed at lower frequencies when the size of the segment being replaced is large. This makes sense as distant cut sites should less frequently come into close proximity to allow NHEJ between them. There is currently no evidence to suggest the general repair machinery can play a role in bringing distant cut sites together. Pax5 positive maturing B cells undergoing V(D)J recombination do not show a proximity bias when recombining V/D or V/J genes within Ig loci using RAG dependent recombination. Long distance recombination events in this case are facilitated by reorganization of the chromosome into ‘V gene loops’ or ‘recombination centers’ which bring V/D and V/J switch sites into proximity during this stage of B cell development (David Schatz, Nature Reviews Immunology, 11, 251-263 2011).

Main text, ninth paragraph and Figure 3B. Was equivalent sequencing performed for the light chains of cells that were not exposed to selection by WITO and MGRM8? This should be shown for comparison with W1,2,3 and M1,2,3. Also do positions 95 and 97 fall within AID hotspots?

Indeed, sequencing on the PU ‘passaged unselected’ cell line was also performed. Because no FACS selection pressure was applied to this line after the original C108 selection for engineered cells, the light chain did not show an enrichment of mutations deleting or moving the N95 and the sequences were very similar to the non-engineered Ramos cell line for the light chain. Results from this sample have now been included in the bar graph in Figure 3B. The accepted AID hotspot motif is WRCY, where W= A or T; R= A or G; and Y= C or T. The CDRL3 indeed has an AID hotspot motif spanning residue 97. We have now covered this in the text of the revised manuscript.

Figure 3B does not make it easy to see which of the mutations result in the shift or deletion of the glycan at position 95 which is the focus of the following section.

We have simplified this figure and now categorize mutations in the CDRL3 into three phenotypes; 1) N95, 2) N97 and 3) N95 deletion. We hope this improves the clarity of the result.

Main text ninth paragraph. The authors state that antibodies with the N95 glycan deleted or shifted to 97 showed improved affinity for HIV SOSIPS. While this seems generally true, it is not true for WITO, one of the selecting antigens, where loss of the glycan results in a drop in affinity. Since WITO-selected cells lose the glycan as often as they shift it, how do the authors explain the selection of deleterious mutations at this level?

The size exclusion profiles of the three variant antibodies expressed in 293F cells (Figure 1—figure supplement 2A) suggest the possibility that the PG9HC/RamosLLC chimeric antibody may be more stable without a PNGS in the CDRL3 loop based on the fact that the variant with the glycan deletion showed a smaller aggregate peak (around 8-14 ml) compared to those with glycans at 95 and 97. While relative amounts of these three antibodies expressed by 293 cells appear similar, it may also be that the presence of a CDRL3 glycan impairs pairing with the PG9 heavy chain in Ramos cells resulting in reduced levels of cell surface expressed BCR. If this is true, it may be that higher levels of glycan deleted BCR on the cell surface compensate for the slight decrease in affinity (for WITO) and allow these cells to end up in our FACS gate despite our attempts to normalize for BCR expression levels using an anti-λ FITC (Figure 3—figure supplement 2). These antibodies may not have been selected if we set a lower anti-λ (FITC) cut-off. Interestingly, while the affinity of this antibody seems to decrease slightly (on the level of on rate), neutralization by this variant is better than for the N95 against WITO. It may be that a slower off rate plays a role in selecting this variant during FACS and in neutralizing the WITO virus.

Figure 3E – how similar are the Ramos and PG9 light chains? Would homology modelling with the correct light chain provide additional confirmation?

The Ramos and PG9 λ light chains use the same V gene (IGLV2-14*01). Ramos uses the IGLJ2*01 gene while PG9 uses the IGLJ3*02 (Figure 1). CDR1 and 2 loops are almost identical in size and sequence. The CDRL3s are significantly different in sequence. The Ramos CDRL3 is one residue longer (11 amino acids) and of course contains the PNGS at position 95. Because of this glycan’s proximity to the N160 glycan of the HIV Env epitope, it is easy to imagine how it may impede approach and binding of the PG9 HC due to steric hinderance. Removal or shifting of its position to reduce this clash could thus improve affinity. WITO, which is less glycan dense than MGRM8 at the apex (missing glycans both within the hypervariable V2 loop (185-187) and at position 130, both in close proximity to the N160 glycan, Voss et al. Cell Reports Figure 1A and B), could mean that the epitope on WITO better accommodates a glycan projecting down towards the epitope from the CDRL3. Atomic resolution structures could clarify matters here, but we feel this is beyond the scope of this report.

Figure 3F – related to the comment above, a similar comparison between PU, M3 and W3 would be informative, given the substantial SHM in the H chain in PU, some of which overlaps that of the antigen selected SHM in M3.

In this experiment, we only wanted to prove that somatic hypermutation was occurring in the new PG9 VDJ gene. We therefore repeated the RT-PCR using a protocol which incorporates unique molecular identifiers (UMIs) into each mRNA as it is transcribed to DNA. Then we amplified by PCR and chose two samples to fill a MiSeq chip (25m reads total) in order to generate enough reads to achieve an average UMI frequency of 30 reads/UMI to deal with PCR based artifacts which complicate AID mutation analysis. We chose the unselected line and one selected line to illustrate how mutations in these genes may become skewed as a result of selection pressure acting on the final protein. This figure proves that AID mutations occur, and that selected genes do show differences compared with unselected ones. We do not think that significant conclusions can be made about the details of this selection in this specific example. For example, one could argue that the presence of a green bar in the coding mutation graph where no purple bar exists means that the mutation results in a BCR which shows lower affinity to the FACS probe which is therefore not selected. But at such low frequency, another option is that a founder effect is occurring during the antigen selection bottleneck whereby this particular mutation is reduced in frequency by chance, especially when very small numbers of cells are selected from the total. Similarly, a purple bar which is higher than a green bar suggests a site where positive selection pressure (improved FACS probe binding) is enriching a mutation. But these could also be founder effects during population bottlenecks during the sort, emergence of mutations after selection, or genetic linkages to other sites which are undergoing purifying selection for example.